# Replay-enhanced Continual Reinforcement Learning

**Tiantian Zhang**[*]                                           *ztt19@mails.tsinghua.edu.cn*
*Tsinghua University*

**Kevin Z. Shen**[*]                                              *kevins00@student.ubc.ca*
*The University of British Columbia*

**Zichuan Lin**                                                 *zichuanlin@tencent.com*
*Tencent*

**Bo Yuan**                                                      *boyuan@ieee.org*
*Tsinghua University*

**Xueqian Wang**                                        *wang.xq@sz.tsinghua.edu.cn*
*Tsinghua University*

**Xiu Li**[†]                                                *li.xiu@sz.tsinghua.edu.cn*
*Tsinghua University*

**Deheng Ye**[†]                                              *dericye@tencent.com*
*Tencent*

**Reviewed on OpenReview:** *https://openreview.net/forum?id=91hfMEUukm*

## Abstract

Replaying past experiences has proven to be a highly effective approach for averting catastrophic forgetting in supervised continual learning. However, some crucial factors are still largely ignored, making it vulnerable to serious failure, when used as a solution to forgetting in continual reinforcement learning, even in the context of perfect memory where all data of previous tasks are accessible in the current task. On the one hand, since most reinforcement learning algorithms are not invariant to the reward scale, the previously well-learned tasks (with high rewards) may appear to be more salient to the current learning process than the current task (with small initial rewards). This causes the agent to concentrate on those salient tasks at the expense of generality on the current task. On the other hand, offline learning on replayed tasks while learning a new task may induce a distributional shift between the dataset and the learned policy on old tasks, resulting in forgetting. In this paper, we introduce RECALL, a replay-enhanced method that greatly improves the plasticity of existing replay-based methods on new tasks while effectively avoiding the recurrence of catastrophic forgetting in continual reinforcement learning. RECALL leverages adaptive normalization on approximate targets and policy distillation on old tasks to enhance generality and stability, respectively. Extensive experiments on the Continual World benchmark show that RECALL performs significantly better than purely perfect memory replay, and achieves comparable or better overall performance against state-of-the-art continual learning methods.

## 1 Introduction

Continual learning, an emerging machine learning paradigm, examines multiple learning tasks in sequence, where the data distribution and learning objective change through time and is considered an important

---

[*]This work was done when Tiantian Zhang and Kevin Z. Shen worked as interns in Tencent.
[†]Corresponding authors.

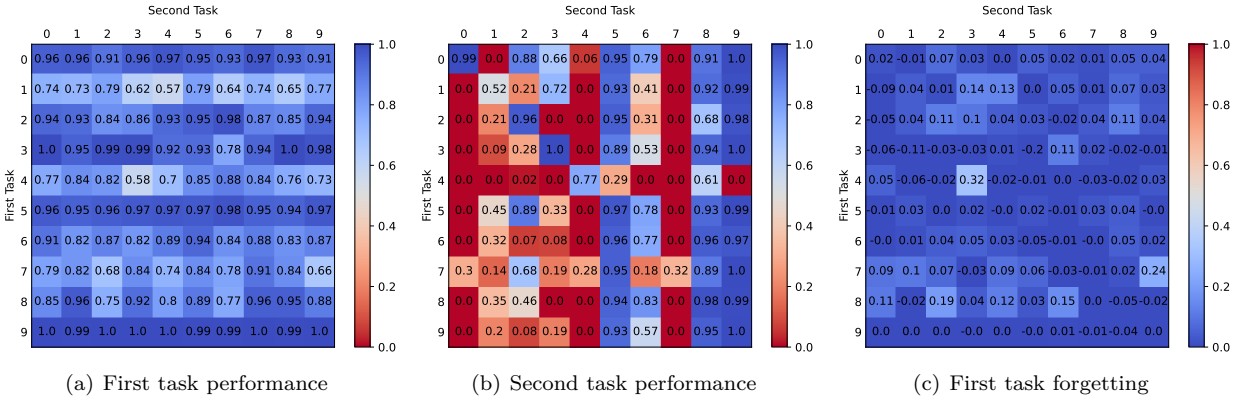

Figure 1: The evaluation matrices in terms of success rate with Perfect Memory on pairwise sequential tasks from Continual World. The numbers $0 \sim 9$ indicate identifications of ten different tasks, and the mapping between them and the proper task names are shown in Figure 7 in Appendix A. For example, if the identifications of two tasks is 5 and 0 respectively, it means that the learning is conducted on task sequence $\mathcal{M} = $ [HANDLE-PRESS-SIDE-V1, HAMMER-V1]. We use the same colorbars to visualize the performance in (a) and (b) and a reversed version to show the level of forgetting in (c), where darker red indicates worse results. The average values of (a), (b), and (c) are 0.87, 0.44 and 0.02, respectively. It is clear that naive experience replay with perfect memory can guarantee the stability to a large extent on RL tasks. Nevertheless, it still exhibits a certain degree of forgetting on some tasks. Even worse, it suffers from severe plasticity restriction on the learning of new tasks.

step toward artificial general intelligence (Parisi et al., 2019; De Lange et al., 2021; Wang et al., 2023). An effective continual learning system must emphasize two potentially conflicting optimization goals. First, when a learned scenario is encountered again, the agent is expected to immediately demonstrate good performance, ideally as good as before. Second, when a new scenario arises, the agent should conduct quick learning and gain new skills without being limited by the maintenance of previously acquired skills. These conflicting objectives — adapting to new tasks while maintaining the knowledge of old ones, correspond to the challenge known as the plasticity-stability dilemma in artificial and biological neural systems (Mirzadeh et al., 2020).

Catastrophic forgetting is the quintessential failure mode of continual learning in which the acquisition of new knowledge gradually overwrites old knowledge, resulting in desirable plasticity but limited stability. Inspired by the memory consolidation mechanism of hippocampus replay inside biological systems, replaying previous data is considered a simple yet effective way to mitigate catastrophic forgetting (Rebuffi et al., 2017; Isele & Cosgun, 2018; Rolnick et al., 2019; Korycki & Krawczyk, 2021), and has been widely adopted in supervised continual learning (Rebuffi et al., 2017; Isele & Cosgun, 2018; Korycki & Krawczyk, 2021).

Different from supervised learning with naturally well scaled loss functions (e.g., cross entropy) and stationary training distribution, reinforcement learning (RL) is a goal-oriented online sequential decision-making and learning process (Sutton & Barto, 2018). It involves iteratively interacting with the environment and collecting experiences, typically with the most recently learned policy, and then using these experiences to improve the policy to maximize the reward function. In this process, the distribution of collected experiences is inherently non-stationary, due to the constantly updated policy. Recently, a technique named CLEAR demonstrates the effectiveness of experience replay for reducing catastrophic forgetting in continual RL (Rolnick et al., 2019). However, other related works (Wołczyk et al., 2021; 2022) show that replay-based continual RL methods suffer from rather poor performance on the newly proposed Continual World benchmark. Based on this, we conducted a systematic experimental study on related tasks. As shown in Figure 1, the naive migration of replay-based methods to continual RL may not perform well on learning sequential tasks by a single learning system with limited representation capacities, even in the context of perfect memory where all experiences are kept in the buffer. More specifically, inspired by the stated balancing issue of multiple tasks competing for limited resources of a single learning system in multi-task deep reinforcement learning (Hessel et al., 2019), the saliency of a task for the agent increases with the magnitude and density of the rewards observed in that task, which may differ dramatically across tasks or at various learning phases within the same task. This factor is likely to encourage the agent to focus on tasks that have been learned well in the past instead

of the current task that presents small and sparse initial rewards (suppressing plasticity). Additionally, using past experiences from old tasks as offline data via RL loss to prevent forgetting is a typical offline RL paradigm due to the absence of further interaction. It may cause standard off-policy RL methods to fail due to overestimation of values induced by the distributional shift between the dataset and the learned policy, degrading the well learned performance on previous tasks and resulting in forgetting (disrupting stability).

In this paper, we address the aforementioned two issues to provide an effective replay-enhanced method for continual RL settings. We propose Replay-Enhanced ContinuAL rL (RECALL), an improved version of the naive replay for continual RL, which incorporates the adaptive normalization mechanism on approximate targets used in value function learning and the policy distillation technique for offline policy preservation. The main contributions of this work are summarized as below:

- **Scale invariant replay-enhanced continual RL.** We investigate the issue of limited plasticity for subsequent tasks in replay-based continual RL settings, and introduce adaptive normalization on the targets to balance the contribution of each task to the agent's updates, alleviating this limitation.

- **Policy distillation for offline policy preservation.** We apply the distillation technique to the policies for old tasks to prevent forgetting caused by offline training, further enhancing stability.

- **Empirical validation on Continual World.** Extensive experiments on a suite of realistic robotic manipulation tasks are conducted to validate the overall superiority of our method over baselines in terms of average performance, forgetting, and forward transfer.

## 2 Related Work

Catastrophic forgetting has long been recognized as a key issue in neural networks, particularly in situations where sequential tasks are learned continuously (Ring, 1997; French, 1999). Recently, a variety of approaches have been investigated to combat catastrophic forgetting in continual learning. According to how the knowledge of previous tasks is retained and leveraged, they can be classified into three major categories: parameter isolation methods, regularization-based methods, and replay methods.

**Parameter isolation methods** This family of works separately optimizes an isolated parameter subspace dedicated to each task throughout the network, where the architectural resources can be fixed (Fernando et al., 2017; Mallya & Lazebnik, 2018) or incrementally expanded (such as the network capacity (Rusu et al., 2016b) or a policy library (Wang et al., 2019; 2022)). These strategies avoid catastrophic forgetting by protecting all weights for the previous tasks from being perturbed by new information but knowledge transfer and generalization between tasks might be restricted, with unnecessary redundancy in the network structure.

**Regularization-based methods** Regularization-based approaches protect learned knowledge from forgetting by imposing an extra regularization term on the learning objective, penalizing large updates on important weights (Kirkpatrick et al., 2017; Kessler et al., 2020) or policies (Rusu et al., 2016a; Traoré et al., 2019; Zhang et al., 2022; 2023) for previous tasks. This family of works requires careful design of regularization terms and fine-tuning of their associated coefficients. It is easy to implement and tends to perform well on small sets of tasks, but still faces performance trade-offs on new and old tasks as their number increases.

**Replay methods** Experience replay is a basic and powerful strategy for reinforcing the significance of experiences from past tasks during continual learning. The core idea of replay methods is to store samples of past tasks (Isele & Cosgun, 2018; Rolnick et al., 2019; Riemer et al., 2019; Korycki & Krawczyk, 2021) or generate pseudo-samples from a generative model (Shin et al., 2017; Atkinson et al., 2021) to maintain knowledge about the past in the network. These previous task samples are replayed while learning new tasks in the form of either being reused as model inputs for rehearsal (Shin et al., 2017; Isele & Cosgun, 2018; Rolnick et al., 2019; Korycki & Krawczyk, 2021; Atkinson et al., 2021) or constraining the optimization of new tasks (Rolnick et al., 2019; Riemer et al., 2019), yielding decent results against catastrophic forgetting.

While storing past experiences in replay methods can be memory-intensive, it is an attractive strategy when memory is sufficient due to its simplicity and excellent performance in reducing forgetting. A theoretical

analysis (Knoblauch et al., 2020) has demonstrated the necessity of perfect memory to resolve the NP-hard problem of optimal continual learning. It also shows that replaying or reconstructing observations from previously observed tasks is likely to be more effective in developing reliable continual learning algorithms in comparison with regularization-based approaches. Meanwhile, some studies (Rebuffi et al., 2017; Isele & Cosgun, 2018; Rolnick et al., 2019) show that it is sufficient to preserve a small quantity of selective experiences using sampling tactics such as reservoir sampling when memory is severely constrained.

Most existing replay-based studies concentrate on classification tasks, whereas only a few works look into deep RL. CLEAR (Rolnick et al., 2019) provides preliminary evidence on the value of replay within the deep RL framework, but it has only been empirically validated on tasks with comparable reward scales, without any consideration of how the scale of rewards across sequential tasks may affect the learning process. Recent works (Wołczyk et al., 2021; 2022) on a benchmark suite for continual RL, called Continual World, indicate that even with perfect memory, common replay-based methods might still suffer from significant failures on certain robotic tasks. By contrast, our proposed RECALL seeks to offer an effective remedy to tackle this challenge, inspired by the power of knowledge distillation (Rusu et al., 2016a; Zhang et al., 2022) and adaptive normalization for target scale invariant updates (van Hasselt et al., 2016; Hessel et al., 2019).

## 3  Preliminaries

**Reinforcement Learning**  RL is commonly studied following the MDP framework, which is defined as a tuple $M = \langle \mathcal{S}, \mathcal{A}, \mathcal{P}, \mathcal{R}, \gamma \rangle$, where $\mathcal{S}$ is the set of states; $\mathcal{A}$ is the set of actions; $\mathcal{P} : \mathcal{S} \times \mathcal{A} \times \mathcal{S} \rightarrow [0, 1]$ is the transition probability function; $\mathcal{R} : \mathcal{S} \times \mathcal{A} \times \mathcal{S} \rightarrow \mathbb{R}$ is the reward function, and $\gamma \in [0, 1]$ is the discount factor. At each time step $t \in \mathbb{N}$, the agent moves from $s_t$ to $s_{t+1}$ with probability $p(s_{t+1}|s_t, a_t)$ after it takes action $a_t$, and receives instant reward $r_t$. The goal of RL is to find an optimal policy from experimental trials and relatively simple feedbacks received, enabling the agent to actively interact with the environment to obtain maximum cumulative reward.

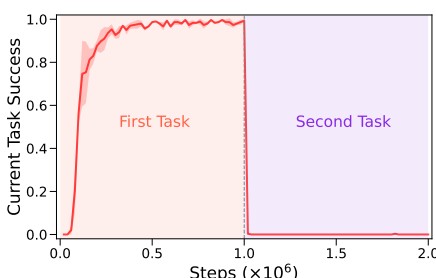

Figure 2: An example of the learning curve of Perfect Memory showing poor plasticity.

**Soft Actor Critic**  Similar to (Wołczyk et al., 2022), we use the soft actor-critic (SAC) (Haarnoja et al., 2018a;b) as the underlying RL algorithm in this paper. It is an off-policy algorithm with experience replay, based on the maximum entropy principle, which is especially beneficial for replay-based continual learning. Formally, let $\pi_\phi(a_t|s_t)$ denote the policy network with parameters $\phi$ and $Q_\theta(s_t, a_t)$ denote the Q-value function with parameters $\theta$. Then, the Q-function can be trained to minimize the soft Bellman residual

$$\mathcal{L}_Q(\theta) = \mathbb{E}_{(s_t, a_t) \sim \mathcal{D}} \left[ \frac{1}{2} \Big( Q_\theta(s_t, a_t) - \big( r(s_t, a_t) + \gamma \mathbb{E}_{s_{t+1} \sim p}[V_{\bar{\theta}}(s_{t+1})] \big) \Big)^2 \right], \tag{1}$$

where $V_{\bar{\theta}}(s_t) = \mathbb{E}_{a_t \sim \pi_\phi} \big[ Q_{\bar{\theta}}(s_t, a_t) - \alpha \log \pi_\phi(a_t|s_t) \big]$ is the soft state value function and $\alpha$ is the temperature parameter that determines the relative importance of the entropy term versus the reward. The policy can be updated by minimizing

$$\mathcal{L}_\pi(\phi) = \mathbb{E}_{s_t \sim \mathcal{D}} \big[ \mathbb{E}_{a_t \sim \pi_\phi}[\alpha \log(\pi_\phi(a_t|s_t)) - Q_\theta(s_t, a_t)] \big]. \tag{2}$$

Notably, under the replay-based continual RL setting, replay buffer $\mathcal{D}$ here stores both new experiences $\mathcal{D}_{new}$ collected from the current task and replayed experiences $\mathcal{D}_{old}$ from the historical ones, i.e., $\mathcal{D} = \mathcal{D}_{new} \cup \mathcal{D}_{old}$.

**Perfect Memory Replay in Continual World**  To examine the replay method in the context of continual RL, we systematically conduct preliminary experiments on 100 sequential tasks created through permuting two of the ten different realistic robotic manipulation tasks (see appendix A) from the latest Continual World (Wołczyk et al., 2021) benchmark, where each task lasts for 1M steps in its corresponding environment. We assume a multi-head network setting, and keep all the experiences in the replay buffer to allow for a

generous replay, dubbed Perfect Memory in (Wołczyk et al., 2021). After ending the training on both tasks, we evaluate the final performance (success rate) on the first and second tasks as well as the forgetting of the first task. The results are shown in Figure 1 from which we can observe the following findings:

- **Decent stability.** According to Figure 1(a) (0.87 average success rate), the agent does perform well on the majority of first tasks after training is complete, which demonstrates that replaying experiences of past tasks, as in supervised learning, can also effectively ensure the stability of continual learning algorithms in RL scenarios.

- **Limited plasticity.** Unfortunately, as shown in Figure 1(b) (0.44 average success rate), the agent shows no (31 out of 100 tasks have a success rate of zero) or weak (25 out of 100 tasks have a success rate of less than 0.5) success on a considerable percentage of the second tasks, indicating that the plasticity is severely restricted. Figure 2 illustrates an example of the training curve in terms of success rate for the current task (the one being trained) that suffers such plasticity limitation, in which the agent does not get any effective learning on the second task. More details about the losses and Q value curves for the corresponding current task are provided in Appendix C.1 (see Figure 8). Additionally, we conducted four experiments in Appendix C.1 that used actor and critic networks that were either shared or not shared among tasks. The results (See Table 5) demonstrate that the limited plasticity suffered by Perfect Memory is primarily due to the shared critic network. Inspired by the studies in (van Hasselt et al., 2016; Hessel et al., 2019), we mainly focus on addressing the adverse effects on value function optimization caused by significant difference in the initially observed reward scale of the subsequent task relative to that of well learned old tasks.

- **Mild forgetting.** While the agent performs well on most of the first tasks, Figure 1(c) (0.02 average forgetting) shows that there are still a few tasks that exhibit some degree of forgetting (the success rate decreased by over 0.1 in 13 out of 100 tasks) and the maximum forgetting reached 0.32. The learning curves for the performance of these 13 tasks are provided in Appendix C.2 (see Figure 9). We also verified that the forgetting shown here is essentially a result of the offline learning for replayed tasks in Appendix C.2 by conducting further experiments of Perfect Memory using offline and online replay modes, respectively (See Table 6).

## 4 The RECALL Method

RECALL employs multi-head neural network training for both actor and critic, with each head being responsible for a specific task, which is widely used in continual learning. We define a task sequence $\mathcal{M} = [M_1, M_2, \ldots, M_N]$ of $N$ tasks, where $M_i, i \in [1, 2, \ldots, N]$ is a specific MDP that symbolizes the $i^{\text{th}}$ task encountered during learning. When the $i^{\text{th}}$ task emerges, the aim of RECALL is to update parameters $\Theta = \{\phi, \theta\}$ of policy $\pi_\phi$ and value function $Q_\theta$ to achieve maximum return on all encountered tasks $[M_1, M_2, \ldots, M_i]$: $\Theta^* = \arg\max_\Theta \sum_{j=1}^{i} J_{M_j}(\Theta)$, where $J_{M_j}$ is the expected return on task $M_j$.

In RECALL, we propose to utilize adaptive normalization on targets to balance the contribution of each task to the agent's updates to ensure the plasticity for new tasks, together with the distillation technique to the policies for old tasks to prevent forgetting caused by offline training. The core components of the training framework are shown in Figure 3.

The PopArt method is a technique specifically used to address the value function scaling problem in deep RL (van Hasselt et al., 2016). It scales the value function to ensure that its output is within a suitable range, which helps to improve training stability and efficiency. In multi-task RL, agents typically switch between different tasks, hence the need to train different value functions for each task. Since each task may have different reward signals, the output range of each task's value function may also differ. This leads to the value function scaling problem (Hessel et al., 2019). Similarly, in replay-based continual RL, value functions on both current and past task experiences need to be learned while learning a new task. Since each task also generally have different reward scales, we can view this process as a multi-task learning on current and historical tasks. Therefore, applying the PopArt method to address the value function scaling problem in replay-based continual RL is reasonable and necessary.

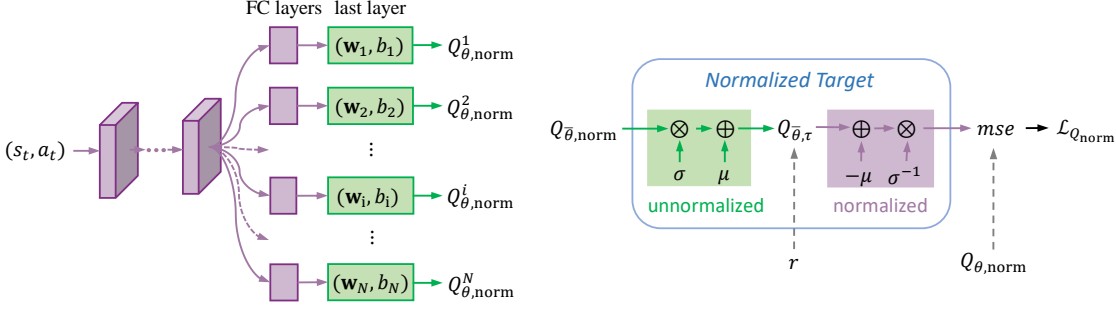

(a) Normalized Q network

(b) Loss function of normalized Q network

Figure 3: The core components of the RECALL scheme. For each input $(s_t, a_t)$, the normalized Q network ultimately outputs only the normalized Q value of the head associated with the task to which it belongs.

To this end, we employ PopArt normalization, developed to derive a scale invariant algorithm for value-based RL, to facilitate learning on new tasks. Concretely, we consider optimizing a normalized value function $Q_{\theta,\text{norm}} = [Q^1_{\theta,\text{norm}}, \ldots, Q^i_{\theta,\text{norm}}, \ldots, Q^N_{\theta,\text{norm}}]$ with $N$ output heads, one for each task in the task sequence. In the following content, for each input $(s_t, a_t)$, we default to using the normalized Q value of the head corresponding to the task to which it belongs and updating the related parameters. Based on this, we omit the subscript $i$ for clarity. Given the targets denoted as $Q_{\bar{\theta},\tau}$, we conduct an affine transformation on it to get normalized targets as $\widetilde{Q}_{\bar{\theta},\tau} = \sigma^{-1}(Q_{\bar{\theta},\tau} - \mu)$, where $\sigma$ and $\mu$ are scale and shift parameters. Notably, in the normalized Q network, each head has its own $(\sigma, \mu)$ learned from the data of the associated task. Under this setting, the loss of $Q_{\theta,\text{norm}}$ can be expressed as:

$$\mathcal{L}_{Q_{\text{norm}}}(\theta) = \mathbb{E}_{(s_t,a_t)\sim\mathcal{D}_{new}\cup\mathcal{D}_{old}}\left[\frac{1}{2}\big(Q_{\theta,\text{norm}}(s_t,a_t) - \widetilde{Q}_{\bar{\theta},\tau}(s_t,a_t)\big)^2\right], \tag{3}$$

where

$$Q_{\bar{\theta},\tau}(s_t,a_t) = r(s_t,a_t) + \gamma\mathbb{E}_{s_{t+1}\sim p}\big[\mathbb{E}_{a_{t+1}\sim\pi_\phi}[\sigma Q_{\bar{\theta},\text{norm}}(s_{t+1},a_{t+1}) + \mu - \alpha\log\pi_\phi(a_{t+1}|s_{t+1})]\big], \tag{4}$$

and $\sigma Q_{\bar{\theta},\text{norm}}(s_t,a_t) + \mu$ is the unnormalized function of the target normalized Q network $Q_{\bar{\theta},\text{norm}}$. Accordingly, the loss function of the policy network is rewritten as:

$$\mathcal{L}_{\pi,\text{norm}}(\phi) = \mathbb{E}_{s_t\sim\mathcal{D}_{new}\cup\mathcal{D}_{old}}\big[\mathbb{E}_{a_t\sim\pi_\phi}[\alpha\log(\pi_\phi(a_t|s_t)) - Q_{\theta,\text{norm}}(s_t,a_t)]\big]. \tag{5}$$

Here, the loss functions $\mathcal{L}_{Q_{\text{norm}}}(\theta)$ and $\mathcal{L}_{\pi,\text{norm}}(\phi)$ are applied on experiences from both old and new tasks. In general, our experiments use a 50-50 experience mixture of novel and replayed tasks, as recommended in (Rolnick et al., 2019). For each sample, only the head associated to the task that it belongs to in the value and policy networks are updated. In addition, after each SAC update, RECALL is required to incrementally update the scale and shift parameters to achieve adaptively targets rescaling:

$$\mu_t = \mu_{t-1} + \beta_t(Q_{\bar{\theta},\tau} - \mu_{t-1}) \quad \text{and} \quad \sigma_t^2 = \nu_t - \mu_t^2, \quad \text{where} \quad \nu_t = \nu_{t-1} + \beta_t(Q_{\bar{\theta},\tau}^2 - \nu_{t-1}), \tag{6}$$

where $\nu_t$ estimates the second moment of the targets, and $\beta_t \in [0,1]$ is the step size. Then, the last layer weights $(\mathbf{w}, b)$ of the corresponding head in the normalized Q network also need to be updated accordingly to preserve the outputs of the unnormalized function precisely after the scale and shift change:

$$\mathbf{w}' = \sigma'^{-1}\sigma\mathbf{w}, \quad b' = \sigma'^{-1}(\sigma b + \mu - \mu'). \tag{7}$$

In addition, in order to prevent forgetting caused by offline training, we employ the policy distillation technique on the replayed tasks to preventing the distributional shift between the past experiences and the learned policies of old tasks while learning a new task. Specifically, the data collection depends only on the policy used for interaction rather than the value function, and the agent generally achieves a good policy on the corresponding task at the end of each task's learning period. Therefore, we only need to add an additional

---

**Algorithm 1** Replay-Enhanced ContinuAL rL (RECALL)

---

**Input**: task sequence $\mathcal{M} = [M_1, M_2, \ldots, M_N]$, policy $\pi_\phi$, value function $Q_\theta$, replay buffer $\mathcal{D}_{old} = \mathcal{D}_{new} = \emptyset$
**Parameter**: regularization coefficient for policy distillation $\lambda$
**Output**: approximate optimal policy and value function $\pi_\phi^*$, $Q_\theta^*$

 1: Train SAC with PopArt normalization on task $M_1$:
 2:      Interact with environment of task $M_1$ and store transitions in $\mathcal{D}_{new}$
 3:      Sample mini-batches from $\mathcal{D}_{new}$ and minimize $\mathcal{L}_{Q_{\mathrm{norm}}}(\theta)$, $\mathcal{L}_{\pi,\mathrm{norm}}(\phi)$.
 4: **for** task $M_i$, $i = 2, .., N$ **do**
 5:      Gather actor outputs $\pi_{old}(\cdot|s_t)$ for each state $s_t \sim \mathcal{D}_{new}$ and populate $\mathcal{D}_{old}$
 6:      $\mathcal{D}_{new} \leftarrow \emptyset$
 7:      Interact with environment of task $M_i$ through the best-return exploration at the beginning of each
        task and store transitions in $\mathcal{D}_{new}$
 8:      Train SAC on task $M_i$, with the following modified update rule:
 9:        Sample $[s_t, a_t, r_t, s_{t+1}] \sim \mathcal{D}_{old} \cup \mathcal{D}_{new}$ and compute $\mathcal{L}_{Q_{\mathrm{norm}}}(\theta)$, $\mathcal{L}_{\pi,\mathrm{norm}}(\phi)$
10:        Sample $[s_t, \pi_{old}(\cdot|s_t)] \sim \mathcal{D}_{old}$ and compute $\mathcal{L}_{\pi_{distill}}(\phi)$
11:        Minimize $\mathcal{L}_{Q_{\mathrm{norm}}}(\theta) + \mathcal{L}_{\pi,\mathrm{norm}}(\phi) + \lambda \mathcal{L}_{\pi_{distill}}(\phi)$.
12: **end for**
13: **return** $\pi_\phi^*$, $Q_\theta^*$.

---

regularization term to the loss for policy (actor) network optimization to penalize the KL divergence between the historical and current policy distributions on the old tasks when training the policy network. Formally, this corresponds to adding the distillation loss function:

$$\mathcal{L}_{\pi_{distill}}(\phi) = \mathbb{E}_{s_t \sim \mathcal{D}_{old}}\big[KL[\pi_\phi(\cdot|s_t), \pi_{old}(\cdot|s_t)]\big]. \tag{8}$$

Note that $\mathcal{L}_{\pi_{distill}}(\phi)$ is only applied on replayed experiences of old tasks, and $\pi_{old}$ is the historical policy obtained after ending the training on the associated replayed task. In our implementations, before each new task training starts, we compute $\pi_{old}(\cdot|s_t)$ for all experiences of the previous task through the latest learned policy and store them along with the corresponding experiences in $\mathcal{D}_{old}$ for subsequent use.

**The Complete Scheme** Finally, we combine Equations 3, 5, and 8 to form a joint optimization scheme. Namely, we solve the continual RL problem based on the experience replay method with the following optimization objective:

$$\min_{\theta,\phi} \mathcal{L}_{Q_{\mathrm{norm}}}(\theta) + \mathcal{L}_{\pi,\mathrm{norm}}(\phi) + \lambda \mathcal{L}_{\pi_{distill}}(\phi) \tag{9}$$

where the hyperparameter $\lambda$ is the policy distillation regularization coefficient to control the deviation degree between the historical and current policy distributions of old tasks. The complete procedure of RECALL is described in Algorithm 1. As an additional note, at the beginning of each new task, we initialize the weights of its associated output head in both actor and critic to the already learned head that obtained the best return on that task to facilitate exploration and adaptation. This is referred to as *best-return exploration* in (Wołczyk et al., 2022) and has been shown experimentally to be a non-negligible SAC component for promoting forward transfer.

## 5 Experimental Evaluation

We conduct comprehensive experiments on a suite of realistic robotic manipulation tasks from the Continual World benchmark (De Lange et al., 2021), seeking to answer the overarching questions:

- Q1: Can RECALL eliminate plasticity limitation while increasing stability?

- Q2: Does RECALL achieve better continual reinforcement learning compared with state-of-the-art methods?

- Q3: How do the adaptive normalization and policy distillation mechanisms affect the continual RL performance, respectively?

- Q4: How is RECALL's scalability regarding longer task sequences?

## 5.1 Experimental Settings

**Datasets**  We perform our experiments on the new Continual World benchmark (De Lange et al., 2021) designed as a testbed for evaluating RL agents with respect to challenges incurred by the continual learning paradigm. It consists of ten realistic robotic manipulation tasks. The structure of the observation and action spaces remains the same between tasks, allowing for multi-task learning with a single learning system. For all tasks, the robot must either manipulate one object with a variable goal position, or two objects with a fixed goal position. The observation space is represented as a 12-dimensional vector containing the coordinates of the robot's gripper and relevant objects. The action space is a 4-dimensional vector describing the gripper movement. Reward functions are shaped to make each task solvable and the binary success metric is used to indicate whether the desired goal has been successfully accomplished. The tasks are arranged in sequences and the training on each task lasts for 1M steps. Continual World provides eight triplet sequences of three tasks to allow rapid experimenting, while a longer sequence contains 10 different tasks arranged in a fixed order (called CW10), and CW20 consists of CW10 repeated twice. See Appendix A for more details.

**Baselines**  We evaluate our method in comparison to five standard baselines: (1) *Fine-tuning* is the vanilla continual learning baseline where the model is trained on sequential tasks without any concern of preventing forgetting or facilitating forward transfer. (2) *EWC* (Kirkpatrick et al., 2017) is a classic regularization-based method that uses the Fisher information matrix to approximate the importance of each weight and apply quadratic regularization to network weights to reduce forgetting. (3) *PackNet* (Mallya & Lazebnik, 2018) strictly prevents the performance from deteriorating on the previous tasks by iteratively pruning, retraining, and freezing parts of the network after each task. It is a parameter isolation method, showing good performance on Continual World (Wołczyk et al., 2021). (4) *ClonEx* (Wołczyk et al., 2022) is another regularization-based method combining behavioral cloning and best-return exploration, which demonstrates the best average performance and forward transfer on Continual World. (5) *Perfect Memory* (Wołczyk et al., 2022) is a simple replay method primarily investigated in this paper which keeps all data from past tasks in the SAC's buffer to avoid forgetting. We abbreviate it to PM in our experimental results for simplicity.

**Implementations**  We use an implementation of the underlying RL algorithm SAC (Haarnoja et al., 2018a;b; Zhou et al., 2022) based on (Wołczyk et al., 2021), in which the maximum entropy coefficient $\alpha$ is tuned automatically according to the adjustment rule provided in (Haarnoja et al., 2018b). We follow exactly the same experimental setup (including network structure and hyperparameters) from (Wołczyk et al., 2022) for all baselines and the common settings for RECALL, ensuring fair comparison. The actor and critic are implemented as two separate MLP networks, each with 4 hidden layers of 256 units and assuming the multi-head setting. The difference is that we keep the actor's single-layer head structure consistent with that in (Wołczyk et al., 2022) while designing the critic's output head with 3 hidden layers to avoid introducing too much bias in new tasks during the value function approximation process. The model was trained on each task for 1M steps, and performance was evaluated by testing the current policy on all tasks every 20k steps. The SAC exploration phase takes $10k$ steps. By default, we employ the best-return exploration in RECALL that reuses old policy head to facilitate exploration when the new task begins, as used by ClonEx, as well as inherit the corresponding critic head for faster adaptation. For each task sequence, we search method-specific regularization coefficient $\lambda$ for policy distillation of RECALL in $\{0.01, 0.1, 1, 10, 100\}$, and the final selected value is 10. Replay buffer size is set to be consistent with that in Perfect Memory and batch size is 128. All experiments were conducted with 5 different seeds and we also provide 90% confidence intervals through bootstrapping. More details can be found in Appendix B.

**Metrics**  Following the convention in (Wołczyk et al., 2021), we use average performance, forgetting, and forward transfer across all tasks as the primary metrics for evaluation. Specifically, assume $p_i(t) \in [0, 1]$ as the success rate of task $i$ at time $t$, and that each of the $N$ tasks is trained for $\Delta$ steps, so that (1) the *average*

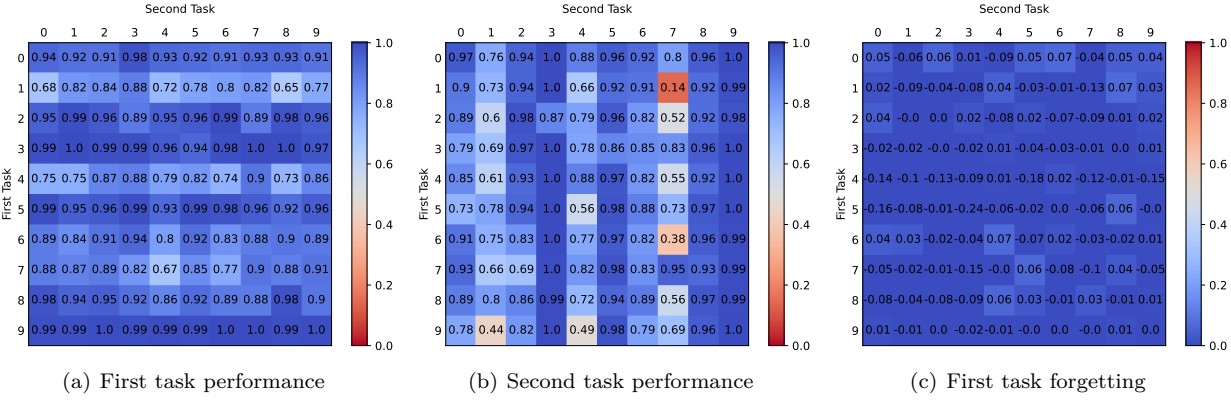

(a) First task performance    (b) Second task performance    (c) First task forgetting

Figure 4: The evaluation matrix for RECALL on pairwise sequential tasks from Continual World. The average values of (a), (b), and (c) are 0.91, 0.85 and −0.02, respectively. It is clear that RECALL considerably enhances the adaptability of the model for new tasks, and also performs well on eliminating mild forgetting, in comparison with Perfect Memory shown in Figure 1.

*performance* at time $t$ is $P(t) = \frac{1}{N} \sum_{i=1}^{N} p_i(t)$; (2) the *forgetting* metric is measured by the average difference between the performance after training on each task versus the performance at the end of training on all tasks, denoted as $F = \frac{1}{N} \sum_{i=1}^{N} p_i(i \cdot \Delta) - p_i(N \cdot \Delta)$; (3) the *forward transfer* for all task is $FT = \frac{1}{N} \sum_{i=1}^{N} FT_i$, where $FT_i$ is the forward transfer of task $i$, defined as a normalized area between its training curve $AUC_i$ and the reference training curve $AUC_i^b$ from training from scratch, i.e., $FT_i = \frac{AUC_i - AUC_i^b}{1 - AUC_i^b}$, $AUC_i = \frac{1}{\Delta} \int_{(i-1) \cdot \Delta}^{i \cdot \Delta} p_i(t) dt$, $AUC_i^b = \frac{1}{\Delta} \int_0^{\Delta} p_i^b(t) dt$, and $p_i^b(t) \in [0, 1]$ is the reference performance.

## 5.2 Plasticity and Stability

Our first experiment was designed to demonstrate the efficacy of RECALL on facilitating plasticity on new tasks as well as reducing forgetting from offline learning (Q1). We apply RECALL to 100 pairs of sequential tasks used in the preceding preliminary experiments and summarize the results in Figure 4. Our method effectively promotes the learning on second tasks, while eliminating mild forgetting caused by offline learning (see Perfect Memory in Figure 1 for reference). More precisely, RECALL reduces the number of second tasks with success rate less than 0.5 to 4 from 56 for Perfect Memory, whilst achieving the dropoff in success rate of less than 0.1 on all 100 first tasks (87 for Perfect Memory). Accordingly, an example of the current task's training curve

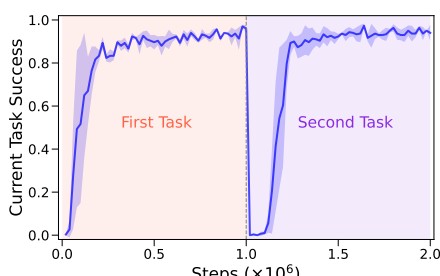

Figure 5: An example of the learning curve of RECALL showing good plasticity.

of RECALL is provided in Figure 5. When the task switches, the agent can quickly adapt to the new environment, showing significantly better plasticity for new tasks than Perfect Memory.

The fact that RECALL can achieve plasticity and stability simultaneously appears to go against the conventional wisdom about the plasticity-stability trade-off, which maintains that the plasticity of artificial and biological neural systems is improved at the expense of stability, whereas too much stability will in turn impede the efficient learning of new knowledge. We argue that the aforementioned perception is fundamentally based on the premise that the capacity of the neural system is fully and well exploited. That is, no additional factors affect the model's performance except for the plasticity-stability dilemma. However, the issue of limited plasticity discussed in this study is caused by the magnitude of rewards rather than excessive attention to stability. Likewise, the mild forgetting that we alleviate is not brought on by too much focus on plasticity, but rather by the offline learning for historical tasks. As a result, it is feasible to address these two parallel issues at the same time to encourage the dual enhancement of plasticity and stability, which is also supported by the results presented in Figure 4.

Table 1: Average performance, forgetting, and forward transfer of all the methods on the triplet sequences (CW3). Here and in related tables, the 90% confidence intervals are provided through bootstrapping. The best possible performance (confidence intervals are considered) for each task is marked in boldface.

**Average Performance**

| CW3 | Fine-tuning | EWC | PackNet | ClonEx | PM | RECALL |
|---|---|---|---|---|---|---|
| 1 | 0.30 [0.29, 0.31] | 0.71 [0.56, 0.85] | 0.70 [0.60, 0.82] | **0.84** [0.77, 0.91] | 0.52 [0.40, 0.59] | **0.90** [0.88, 0.92] |
| 2 | 0.31 [0.29, 0.33] | 0.58 [0.54, 0.62] | 0.86 [0.83, 0.89] | **0.90** [0.79, 0.96] | 0.74 [0.65, 0.83] | **0.92** [0.89, 0.94] |
| 3 | 0.24 [0.22, 0.26] | 0.42 [0.30, 0.51] | 0.61 [0.53, 0.69] | 0.73 [0.64, 0.81] | 0.26 [0.15, 0.36] | **0.91** [0.90, 0.93] |
| 4 | 0.33 [0.32, 0.33] | 0.75 [0.62, 0.89] | 0.53 [0.43, 0.62] | **0.88** [0.83, 0.93] | 0.28 [0.24, 0.32] | **0.87** [0.84, 0.89] |
| 5 | 0.33 [0.33, 0.33] | 0.54 [0.45, 0.62] | 0.74 [0.65, 0.84] | **0.89** [0.79, 0.97] | 0.35 [0.28, 0.46] | **0.92** [0.90, 0.94] |
| 6 | 0.27 [0.24, 0.29] | 0.82 [0.74, 0.89] | 0.40 [0.32, 0.49] | 0.74 [0.69, 0.80] | 0.33 [0.25, 0.45] | **0.91** [0.89, 0.93] |
| 7 | 0.33 [0.33, 0.33] | 0.80 [0.68, 0.92] | 0.91 [0.86, 0.95] | **0.90** [0.81, 0.99] | 0.90 [0.78, 0.97] | **0.95** [0.93, 0.96] |
| 8 | 0.33 [0.33, 0.33] | 0.41 [0.34, 0.52] | 0.81 [0.67, 0.93] | 0.81 [0.62, 0.93] | 0.50 [0.40, 0.61] | **0.95** [0.93, 0.97] |
| mean | 0.31 | 0.63 | 0.70 | 0.84 | 0.49 | **0.92** |

**Forgetting**

| CW3 | Fine-tuning | EWC | PackNet | ClonEx | PM | RECALL |
|---|---|---|---|---|---|---|
| 1 | 0.59 [0.58, 0.61] | 0.10 [-0.02, 0.22] | 0.02 [0.00, 0.04] | 0.02 [-0.01, 0.07] | 0.02 [0.01, 0.04] | −0.01 [-0.03, 0.02] |
| 2 | 0.53 [0.51, 0.55] | 0.26 [0.23, 0.29] | −0.05 [-0.09, 0.00] | −0.04 [-0.08, -0.01] | 0.04 [-0.04, 0.14] | 0.00 [-0.02, 0.02] |
| 3 | 0.61 [0.59, 0.63] | 0.24 [0.21, 0.27] | −0.06 [-0.13, 0.01] | 0.01 [-0.06, 0.09] | −0.02 [-0.07, 0.02] | −0.04 [-0.06, -0.01] |
| 4 | 0.53 [0.51, 0.54] | −0.03 [-0.10, 0.07] | −0.06 [-0.09, -0.02] | −0.03 [-0.09, 0.04] | 0.02 [-0.01, 0.05] | 0.01 [-0.01, 0.03] |
| 5 | 0.55 [0.49, 0.59] | 0.03 [-0.01, 0.07] | 0.01 [-0.05, 0.07] | −0.01 [-0.05, 0.03] | 0.01 [0.00, 0.02] | −0.03 [-0.05, -0.02] |
| 6 | 0.58 [0.56, 0.60] | 0.04 [-0.03, 0.12] | 0.02 [0.01, 0.03] | 0.05 [-0.02, 0.13] | −0.01 [-0.03, 0.01] | −0.03 [-0.05, 0.00] |
| 7 | 0.55 [0.51, 0.58] | 0.13 [0.00, 0.27] | 0.02 [-0.01, 0.05] | 0.04 [-0.04, 0.12] | −0.01 [-0.03, 0.00] | −0.01 [-0.03, 0.01] |
| 8 | 0.57 [0.53, 0.61] | 0.16 [0.04, 0.26] | 0.02 [-0.04, 0.08] | 0.05 [-0.01, 0.12] | 0.00 [-0.02, 0.02] | −0.06 [-0.09, -0.03] |
| mean | 0.56 | 0.12 | −0.01 | 0.01 | 0.01 | **−0.02** |

**Forward Transfer**

| CW3 | Fine-tuning | EWC | PackNet | ClonEx | PM | RECALL |
|---|---|---|---|---|---|---|
| 1 | 0.14 [0.05, 0.21] | −0.10 [-0.24, 0.02] | −0.23 [-0.47, 0.03] | **0.32** [0.24, 0.40] | −0.96 [-1.32, -0.62] | **0.35** [0.29, 0.41] |
| 2 | 0.22 [0.11, 0.32] | 0.03 [-0.13, 0.19] | −0.10 [-0.22, 0.01] | 0.32 [0.03, 0.52] | −0.09 [-0.25, 0.09] | **0.47** [0.44, 0.50] |
| 3 | 0.33 [0.29, 0.38] | 0.03 [-0.19, 0.16] | 0.02 [-0.19, 0.17] | 0.31 [0.16, 0.42] | −0.57 [-0.68, -0.47] | **0.49** [0.45, 0.53] |
| 4 | 0.40 [0.36, 0.44] | 0.26 [0.21, 0.29] | −0.13 [-0.34, 0.08] | **0.41** [0.23, 0.53] | −0.29 [-0.37, -0.21] | **0.45** [0.41, 0.48] |
| 5 | **0.51** [0.42, 0.59] | 0.12 [-0.08, 0.31] | 0.23 [0.15, 0.31] | **0.48** [0.32, 0.62] | −0.19 [-0.33, -0.07] | **0.52** [0.46, 0.57] |
| 6 | 0.31 [0.18, 0.42] | 0.19 [-0.02, 0.35] | −0.15 [-0.48, 0.08] | 0.41 [0.26, 0.53] | −0.70 [-1.02, -0.42] | **0.55** [0.52, 0.58] |
| 7 | 0.32 [0.27, 0.38] | 0.28 [0.20, 0.36] | 0.08 [-0.01, 0.17] | **0.56** [0.50, 0.63] | −0.10 [-0.43, 0.14] | **0.55** [0.48, 0.62] |
| 8 | **0.47** [0.39, 0.54] | −0.09 [-0.42, 0.16] | −0.01 [-0.50, 0.38] | 0.30 [-0.81, 0.17] | −0.44 [-0.78, -0.14] | **0.52** [0.47, 0.55] |
| mean | 0.34 | 0.09 | −0.03 | 0.39 | −0.42 | **0.49** |

## 5.3 Performance Evaluation

Here we systematically perform a quantitative evaluation of RECALL against the five standard baseline methods (Fine-tuning, EWC, PackNet, ClonEx, and PM) (Q2). We apply them to eight triplets (referred to as CW3) and their twice repeated version (referred to as CW6) for fast experimenting and summarized the results in Table 1 and Table 2. The networks used in all task sequences are exactly the same. From the results, we find that RECALL obtained slightly better overall performance than ClonEx, the state-of-the-art method, and significantly better performance than the other four baselines, across all three metrics of average performance, forgetting, and forward transfer.

Table 2: Average performance, forgetting, and forward transfer of all the methods on CW6.

**Average Performance**

| CW6 | Fine-tuning | EWC | PackNet | ClonEx | PM | RECALL |
|---|---|---|---|---|---|---|
| 1 | 0.10 [0.06, 0.14] | 0.71 [0.57, 0.84] | 0.79 [0.71, 0.87] | 0.87 [0.81, 0.92] | 0.47 [0.44, 0.50] | **0.95** [0.94, 0.95] |
| 2 | 0.16 [0.16, 0.17] | 0.59 [0.41, 0.74] | 0.80 [0.74, 0.86] | 0.90 [0.85, 0.95] | 0.50 [0.45, 0.55] | **0.98** [0.97, 0.99] |
| 3 | 0.11 [0.09, 0.12] | 0.61 [0.57, 0.65] | 0.50 [0.42, 0.59] | 0.81 [0.75, 0.85] | 0.14 [0.13, 0.14] | **0.87** [0.86, 0.88] |
| 4 | 0.17 [0.17, 0.17] | 0.56 [0.53, 0.58] | **0.86** [0.82, 0.89] | **0.85** [0.81, 0.88] | 0.17 [0.13, 0.21] | **0.89** [0.86, 0.90] |
| 5 | 0.17 [0.17, 0.17] | 0.42 [0.33, 0.52] | 0.75 [0.61, 0.87] | 0.91 [0.86, 0.96] | 0.32 [0.29, 0.37] | **0.97** [0.97, 0.98] |
| 6 | 0.13 [0.13, 0.14] | 0.75 [0.65, 0.85] | 0.64 [0.57, 0.71] | 0.74 [0.70, 0.79] | 0.28 [0.17, 0.39] | **0.95** [0.94, 0.97] |
| 7 | 0.17 [0.17, 0.17] | **0.96** [0.95, 0.96] | 0.87 [0.79, 0.93] | **0.96** [0.94, 0.98] | 0.85 [0.75, 0.95] | **0.97** [0.95, 0.99] |
| 8 | 0.17 [0.17, 0.18] | 0.51 [0.43, 0.59] | 0.82 [0.67, 0.96] | **0.97** [0.96, 0.98] | 0.64 [0.61, 0.65] | **0.95** [0.92, 0.97] |
| mean | 0.15 | 0.64 | 0.75 | 0.88 | 0.42 | **0.94** |

**Forgetting**

| CW6 | Fine-tuning | EWC | PackNet | ClonEx | PM | RECALL |
|---|---|---|---|---|---|---|
| 1 | 0.71 [0.67, 0.75] | 0.07 [0.00, 0.13] | 0.00 [-0.02, 0.02] | 0.02 [-0.01, 0.05] | 0.01 [-0.02, 0.04] | −0.05 [-0.06, -0.04] |
| 2 | 0.73 [0.71, 0.75] | 0.20 [0.11, 0.29] | 0.00 [-0.02, 0.01] | 0.02 [-0.02, 0.06] | 0.05 [0.00, 0.11] | −0.05 [-0.06, -0.03] |
| 3 | 0.70 [0.68, 0.72] | 0.02 [-0.05, 0.08] | −0.03 [-0.07, -0.01] | 0.04 [0.01, 0.08] | 0.00 [-0.01, 0.01] | −0.05 [-0.07, -0.04] |
| 4 | 0.59 [0.54, 0.64] | 0.05 [0.03, 0.06] | −0.04 [-0.07, 0.00] | 0.02 [-0.01, 0.06] | −0.02 [-0.05, 0.01] | −0.04 [-0.06, -0.02] |
| 5 | 0.74 [0.70, 0.77] | 0.01 [-0.01, 0.02] | −0.04 [-0.08, -0.01] | 0.03 [-0.02, 0.08] | −0.01 [-0.04, 0.02] | −0.04 [-0.05, -0.04] |
| 6 | 0.68 [0.62, 0.74] | 0.01 [-0.03, 0.06] | −0.01 [-0.03, 0.00] | 0.08 [0.02, 0.15] | −0.02 [-0.05, 0.01] | −0.03 [-0.03, -0.02] |
| 7 | 0.75 [0.73, 0.78] | −0.06 [-0.09, -0.03] | −0.03 [-0.08, 0.02] | −0.01 [-0.03, 0.01] | 0.07 [0.00, 0.15] | −0.01 [-0.03, 0.00] |
| 8 | 0.71 [0.64, 0.75] | 0.05 [-0.01, 0.11] | −0.03 [-0.06, 0.00] | −0.01 [-0.03, 0.01] | 0.01 [0.00, 0.02] | −0.05 [-0.06, -0.03] |
| mean | 0.70 | 0.04 | −0.02 | 0.02 | 0.01 | **−0.04** |

**Forward Transfer**

| CW6 | Fine-tuning | EWC | PackNet | ClonEx | PM | RECALL |
|---|---|---|---|---|---|---|
| 1 | 0.00 [-0.11, 0.11] | −0.02 [-0.18, 0.14] | −0.09 [-0.19, 0.00] | **0.34** [0.22, 0.45] | −0.92 [-1.04, -0.82] | **0.36** [0.32, 0.40] |
| 2 | 0.26 [0.19, 0.33] | −0.02 [-0.27, 0.17] | −0.09 [-0.20, 0.02] | **0.51** [0.45, 0.57] | −0.65 [-0.90, -0.42] | **0.55** [0.50, 0.59] |
| 3 | 0.24 [0.18, 0.29] | −0.11 [-0.26, 0.03] | −0.08 [-0.19, 0.03] | **0.46** [0.43, 0.49] | −0.66 [-0.74, -0.60] | **0.48** [0.45, 0.50] |
| 4 | 0.28 [0.25, 0.31] | 0.13 [0.05, 0.20] | 0.41 [0.36, 0.45] | **0.54** [0.51, 0.57] | −0.49 [-0.57, -0.42] | **0.53** [0.50, 0.56] |
| 5 | 0.42 [0.30, 0.53] | −0.12 [-0.28, 0.05] | 0.18 [0.02, 0.34] | **0.67** [0.61, 0.72] | −0.25 [-0.34, -0.18] | **0.67** [0.64, 0.70] |
| 6 | 0.39 [0.32, 0.45] | 0.18 [-0.03, 0.34] | 0.02 [-0.15, 0.17] | 0.33 [0.15, 0.51] | −0.73 [-1.07, -0.42] | **0.65** [0.59, 0.70] |
| 7 | 0.45 [0.41, 0.48] | 0.25 [0.12, 0.37] | −0.27 [-0.59, 0.06] | 0.55 [0.48, 0.62] | −0.05 [-0.30, 0.19] | **0.64** [0.59, 0.68] |
| 8 | 0.43 [0.35, 0.50] | −0.20 [-0.42, -0.02] | 0.05 [-0.38, 0.42] | **0.68** [0.62, 0.73] | 0.11 [0.02, 0.18] | **0.61** [0.56, 0.66] |
| mean | 0.31 | 0.01 | 0.02 | 0.51 | −0.46 | **0.56** |

It is worth noting that the fundamental reason that RECALL outperforms ClonEx is that they use completely different mechanisms to alleviate catastrophic forgetting. To be specific, ClonEx is a regularization-based approach that reduces forgetting by adding a regularization term to constrain updates of network weights. In general, if the network capacity is adequate, the optimal outcome that can be attained by this mechanism is to entirely preserve the performance on previous tasks and achieve zero forgetting. It rarely obtains positive backward transfer unless the solution space of subsequent tasks includes that of historical tasks. According to the experimental results, it generally exhibits some level of forgetting on most task sequences due to the requirement to ensure plasticity on following tasks, which is particularly apparent in long task sequences.

By contrast, to avoid catastrophic forgetting, RECALL maintains the training on past tasks by replaying experiences while learning new tasks. If the agent does not reach optimal performance at the end of the

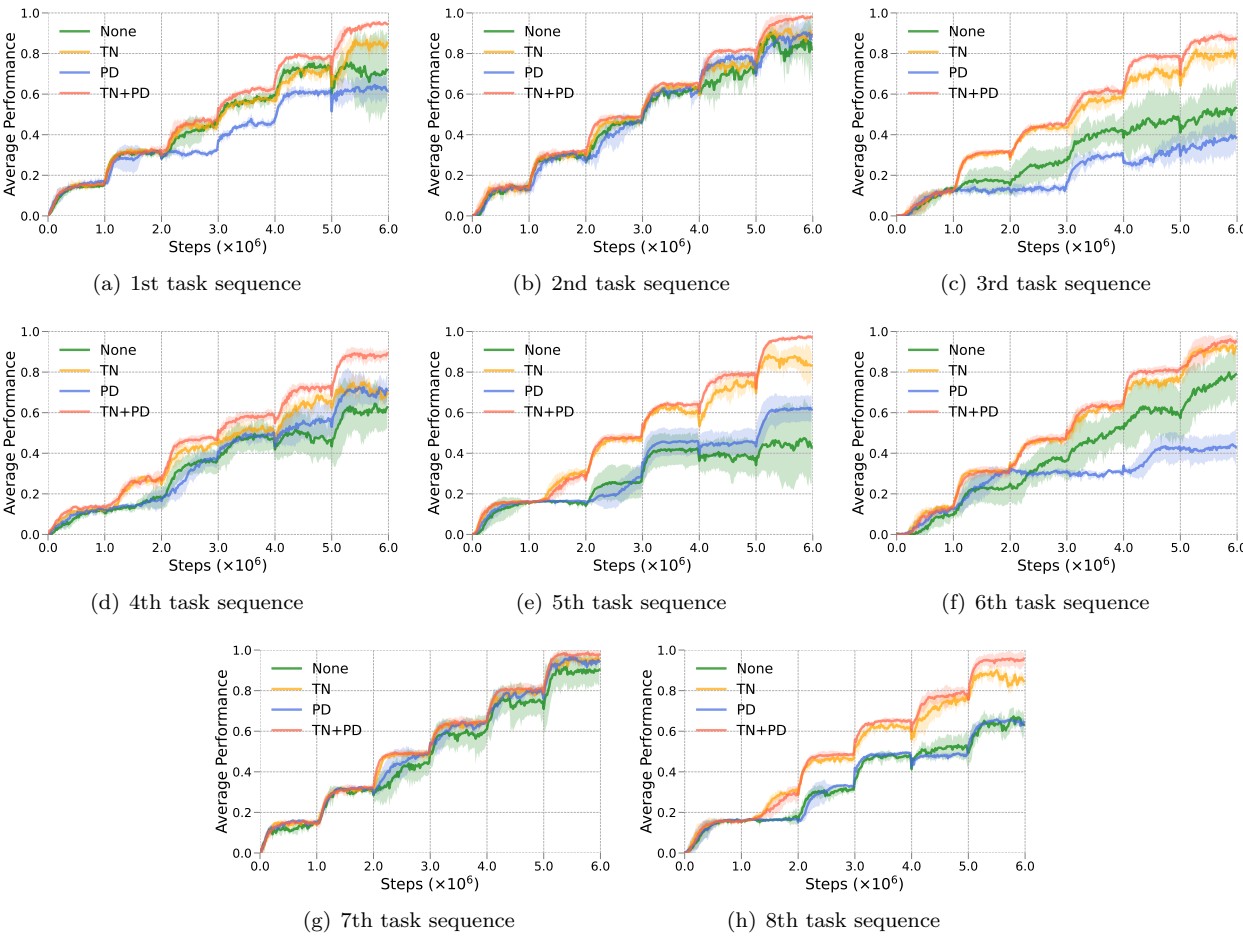

Figure 6: Average (over tasks) success rate per iteration of the four variants in CW6 task sequences.

respective training period of the old tasks, such experience replay (further training) is likely to make the agent perform better on these tasks rather than just preventing catastrophic forgetting. Consequently, RECALL can allow for positive backward transfer (i.e., produce negative forgetting values and improve average performance). Furthermore, the combination of experiences from new and replayed tasks for joint training can aid the model in finding a better common solution space, improving the final performance on all tasks, and also facilitating faster learning of new tasks to achieve more positive forward transfer relative to regularization-based methods.

## 5.4 Ablation Study

In this section, we consider the individual effects of target normalization (TN) and policy distillation (PD) mechanisms (Q3). To this end, we conduct experiments by manipulating a single variable at a time for in-depth analysis. For each new task, the following four variants of the proposed method are applied for continual learning in the new environment: (1) *None:* Neither target normalization nor policy distillation mechanism is used, i.e., degenerating to the naive experience replay with best-return exploration. (2) *TN:* Only apply target normalization mechanism. (3) *PD:* Only apply policy distillation mechanism. (4) *TN+PD:* Both target normalization and policy distillation mechanisms are used, i.e., representing RECALL. The learning curves in terms of average performance per iteration of these four variants on CW6 task sequences are shown in Figure 6.

First, the variants of None and TN (or PD and TN+PD) are compared to identify how the target normalization mechanism affects continual learning performance. In all eight task sequences, the targets are normalized for each task throughout the whole training process, and the performance in terms of average success rate

Table 3: Results of all the methods on CW10 and CW20 sequences. Average performance (Ave. Perf.), forgetting, and forward transfer (F. Transfer) are shown in columns.

| | **CW10** | | | **CW20** | | |
|---|---|---|---|---|---|---|
| Method | Ave. Perf. | Forgetting | F. Transfer | Ave. Perf. | Forgetting | F. Transfer |
| Fine-tuning | 0.10 [0.10, 0.10] | 0.74 [0.72, 0.76] | 0.29 [0.25, 0.32] | 0.05 [0.05, 0.05] | 0.73 [0.69, 0.76] | 0.20 [0.14, 0.26] |
| EWC | 0.61 [0.59, 0.63] | 0.06 [0.05, 0.08] | 0.03 [-0.04, 0.09] | 0.61 [0.55, 0.68] | 0.02 [-0.01, 0.06] | −0.13 [-0.21, -0.04] |
| PackNet | **0.87** [0.83, 0.91] | −0.04 [-0.06, -0.02] | 0.29 [0.22, 0.35] | 0.79 [0.75, 0.82] | −0.01 [-0.03, 0.00] | 0.16 [0.08, 0.22] |
| ClonEx | 0.85 [0.80, 0.90] | 0.00 [-0.02, 0.03] | **0.39** [0.36, 0.43] | 0.82 [0.79, 0.86] | 0.05 [0.04, 0.06] | **0.39** [0.35, 0.42] |
| PM | 0.26 [0.23, 0.28] | 0.02 [-0.01, 0.06] | −1.23 [-1.37, -1.10] | 0.09 [0.03, 0.15] | 0.10 [0.04, 0.16] | −1.36 [-1.44, -1.29] |
| RECALL | **0.89** [0.86, 0.91] | −0.03 [-0.04, -0.03] | **0.40** [0.35, 0.43] | **0.90** [0.87, 0.92] | −0.04 [-0.05, -0.03] | **0.42** [0.39, 0.44] |

is maintained or slightly improved for the first task (the first 1M steps) and dramatically enhanced for the subsequent tasks. It verifies that training a scale invariant value function (a normalized Q network) in replay-based continual RL leads to better adaptation for subsequent tasks, as stated in Section 4.

Next, the PD and TN variants are compared with None and TN+PD to verify the effectiveness of the policy distillation mechanism. Compared with the naive experience replay, distilling the policy from previous tasks when learning on the new task can achieve a certain degree of performance improvement on some task sequences. Conversely, it shows an obvious performance degradation on other sequences, potentially because excessive policy distillation inhibits the learning of new tasks. By contrast, applying the policy distillation to the TN variant can consistently improve the performance of all task sequences. This observation confirms the assumption in Section 4 that applying a proper distillation of the policies from old tasks can help mitigate the mild forgetting caused by offline learning.

Finally, all four variants are compared. It can be observed that the target normalization mechanism can improve learning performance better than policy distillation, and combining the two mechanisms together (TN+PD) leads to the best performance on those various task sequences.

### 5.5   Scalability

To evaluate the scalability of RECALL as well as how it compares with other baseline methods, we test them against longer task sequences CW10 and CW20 (Q4). As shown in Table 3, RECALL outperforms or matches the overall performance of all other baselines on CW10, and is consistently superior to them on CW20 except for the forward transfer which is comparable to ClonEx. One possible reason that PackNet can also perform well in reducing forgetting is that its total training time for each task is longer since after the initial network training, it undergoes iterative pruning, freezing and retraining parts of the network. Nonetheless, this advantage is rather minor and RECALL still surpasses PackNet on average performance on CW20 and is significantly better than it on all task sequences investigated in our experiments in terms of forward transfer. In addition, the performance gap becomes more evident on the longer task sequence CW20 where RECALL outperforms PackNet in all aspects along with the other methods. A possible factor is that PackNet struggles against the increasing complexity of managing shared and task-specific parameters as the number of tasks becomes large. More generally, we find that while all other methods face a considerable drop in performance as the task sequence length doubled from CW10 to CW20, RECALL experiences the opposite with improvements across all three aspects, albeit marginally. These results highlight the desirable scalability of RECALL, and its robustness in handling lengthy task sequences, rendering it a promising solution for continual RL in complicated scenarios.

## 6   Conclusions

In this work, we present a systematic investigation of replay-based continual RL. Due to the potentially significant difference in scale of rewards across tasks contained in the same sequence, we observe that there exists a serious limitation in the plasticity for subsequent tasks, which hinders the learning of new tasks and

further limits the continual RL agent's final performance. To address this, we propose RECALL to optimize a scale invariant normalized value function by introducing an adaptive normalization mechanism on targets, so that all new tasks will have a similar impact on the learning dynamics to that of the previously well-learned tasks, thus allowing the efficient learning of subsequent tasks. In addition, the policy distillation mechanism for old tasks is used to further alleviate forgetting caused by offline learning on the replayed tasks. Extensive experiments on a suite of realistic robotic manipulation task sequences show that RECALL significantly outperforms or matches state-of-the-art baselines in terms of average performance, forgetting, and forward transfer. Meanwhile, it demonstrates superior scalability on longer task sequences.

Note that the two mechanisms contained in RECALL can serve as a plug-and-play component that can be effortlessly integrated into existing replay-based algorithms to improve their performance on continual RL tasks. We believe that this work constitutes the first step towards understanding the difference between experience replay in supervised continual learning and continual RL. Furthermore, it provides a promising prospect for the adoption and extension of replay-based continual learning techniques in the RL context.

**Acknowledgments**

This work was supported by the STI 2030-Major Projects under Grant 2021ZD0201404 and Tencent Rhino-Bird Research Elite Program.

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

## A    Continual World Benchmark

We briefly present the ten robotic tasks from the Continual World benchmark in Figure 7. The details of the task sequences CW3 used in this paper are as follows:

1. PUSH-V1 → WINDOW-CLOSE-V1 → HAMMER-V1

2. HAMMER-V1 → WINDOW-CLOSE-V1 → FAUCET-CLOSE-V1

3. STICK-PULL-V1 → PUSH-BACK-V1 → PUSH-WALL-V1

4. PUSH-WALL-V1 → SHELF-PLACE-V1 → PUSH-BACK-V1

5. FAUCET-CLOSE-V1 → SHELF-PLACE-V1 → PUSH-BACK-V1

6. STICK-PULL-V1 → PEG-UNPLUG-SIDE-V1 → STICK-PULL-V1

7. WINDOW-CLOSE-V1 → HANDLE-PRESS-SIDE-V1 → PEG-UNPLUG-SIDE-V1

8. FAUCET-CLOSE-V1 → SHELF-PLACE-V1 → PEG-UNPLUG-SIDE-V1

CW6 is CW3 repeated twice. The CW10 sequence is:

1. HAMMER-V1 → PUSH-WALL-V1 → FAUCET-CLOSE-V1 → PUSH-BACK-V1 → STICK-PULL-V1 → HANDLE-PRESS-SIDE-V1 → PUSH-V1 → SHELF-PLACE-V1 → WINDOW-CLOSE-V1 → PEG-UNPLUG-SIDE-V1

CW20 is CW10 repeated twice.

## B    Implementation Details

We use the same hyperparameters as (Wołczyk et al., 2022) for the underlying SAC algorithm. Table 4 lists the numerical settings of some core parameters in the experimental evaluation. For the method-specific hyperparameters involved in the continual learning baselines compared in our experiments, we also inherit the final values obtained after tuning in (Wołczyk et al., 2022):

- EWC: selected regularization coefficient for actor is $10,000$ and that for critic is $0$.

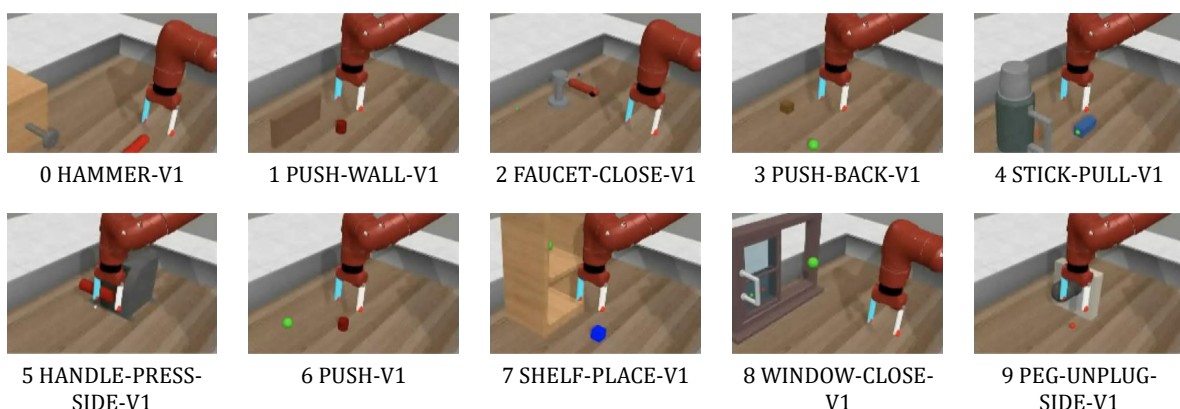

Figure 7: Ten robotic tasks adopted by Continual World benchmark.

Table 4: Core hyperparameters used for the underlying SAC algorithm.

| Parameter | Value |
|---|---|
| optimizer | Adam |
| learning rate | $1 \times 10^{-3}$ |
| batch size | 128 |
| discount factor ($\gamma$) | 0.99 |
| nonlinearity | ReLU |
| target smoothing coefficient ($\tau$) | 0.005 |
| target update interval | 1 |
| target output std ($\sigma_t$) | 0.089 |
| replay buffer size | $10^6$ |

- PackNet: the number of retraining steps is set to $100,000$, and global gradient norm clipping is $2 \times 10^{-5}$.

- ClonEx: selected regularization coefficient for actor is 100 and that for critic is 0. Episodic memory per task is set to $10,000$, and global gradient norm clipping is 0.1.

- Perfect Memory: selected batch size is 512, and replay buffer size is $N \times 10^6$, where $N$ is the number of tasks in the task sequence to be learned.

# C Additional Experimental Results

## C.1 Plasticity Analysis of Perfect Memory

Figure 8 shows the learning curves in terms of actor and critic losses and average predicted action value on the current task, corresponding to the task sequence described in Figure 2). It can be observed that within a few time steps after the learning of a new task (i.e., the second task), the loss functions of actor and critic together with the output of the value function rapidly converge to a stable state that remains around 0, further indicating that the model does not learn any useful information about the new task.

We further investigate the correlation between the above failures and network sharing by experimenting with the actor and critic networks in the following four different combinations:

1) *Shared actor and shared critic:* All tasks adopt the same shared policy network and value network for policy learning.

2) *Unshared actor and shared critic:* All tasks adopt the same shared value network but separate actor network for each task for policy learning.

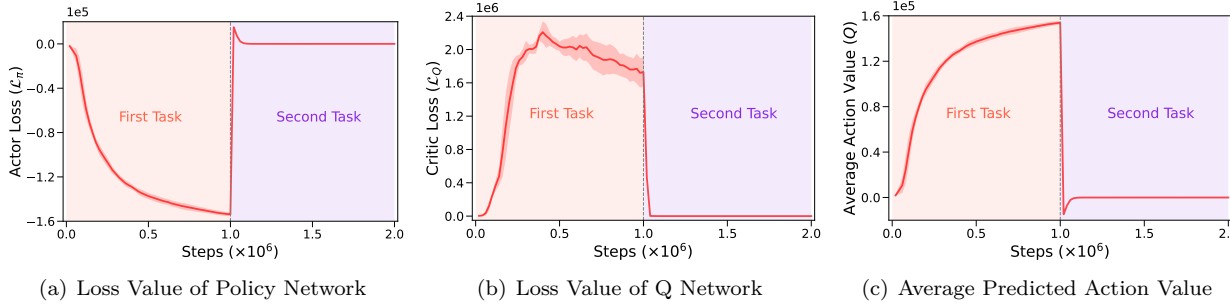

(a) Loss Value of Policy Network     (b) Loss Value of Q Network     (c) Average Predicted Action Value

Figure 8: Learning curves of Perfect Memory with poor plasticity on the current task in terms of actor and critic losses and average predicted action value (corresponding to the task sequence described in Figure 2).

Table 5: Comparison of the final performance (success rate) on each task obtained by Perfect Memory under different actor and critic networks settings (taking the paired task sequence described in Figure 2 as an illustration). Here, "shared" means that two tasks share the corresponding network for policy learning, while "unshared" indicates that a separate network is initialized for each task.

| Network Setting | First Task | Second Task |
|---|---|---|
| shared actor, shared critic | 0.96 [0.94, 0.97] | 0.00 [0.00, 0.00] |
| unshared actor, shared critic | 0.96 [0.95, 0.97] | 0.00 [0.00, 0.00] |
| shared actor, unshared critic | 0.95 [0.93, 0.98] | 0.93 [0.90, 0.95] |
| unshared actor, unshared critic | 0.96 [0.95, 0.98] | 0.94 [0.92, 0.97] |

3) *Shared actor and unshared critic:* All tasks adopt the same shared policy network but separate critic network for each task for policy learning.

4) *Unshared actor and unshared critic:* Building a separate policy network and a separate value network for each task for policy learning. It can be regarded as an upper bound for continuous RL.

Similarly, taking the task sequence described in Figure 2 as an example, the final performance on each task is summarized in Table 5. It can be seen that the limited plasticity of the model on the new task is largely caused by the shared critic network.

## C.2 Forgetting Analysis of Perfect Memory

To further investigate the forgetting performance of Perfect Memory, we visualize the learning curves of the first task in the 13 pairs of task sequences with more than 0.1 forgetting. As seen from Figure 9, all the first tasks achieve relatively good performance after ending learning in their own environment. Nevertheless, as the second task is learned, the performance of the first tasks declines significantly after maintaining for a period of time. That is, in the continual RL setting, forgetting may still occur for experience replay based methods. The magnitude of it is relatively small but non-negligible, so there is still a need to address it further.

We use the following two replay modes to conduct additional experiments of Perfect Memory on 13 task sequences with obvious forgetting in Figure 1 to verify the influence of offline learning on this issue.

- *Offline learning:* The replay mode actually implemented in Perfect Memory. Replaying experiences collected from past tasks while learning the new task, rather than interacting with the old environments again as that is not allowed under continual learning paradigm.

- *Online learning:* Gathers experiences for replayed tasks by interacting with the old environments again while learning the new task. Although this pattern goes against the principle that continual

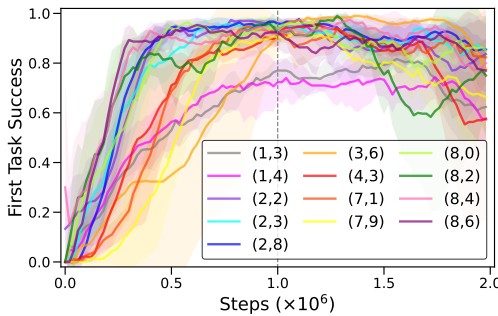

Figure 9: Success rate of the first task for Perfect Memory on the 13 pairs of task sequences with more than 0.1 forgetting. Each curve corresponds to a specific pairwise task sequence. The gray vertical dashed line represents the boundary of task switching.

Table 6: Performance and forgetting of Perfect Memory on the first task of 13 pairwise task sequences under different replay modes.

| Task Sequence | Offline Learning | | Online Learning | |
|---|---|---|---|---|
| | Performance | Forgetting | Performance | Forgetting |
| (1,3) | 0.62 [0.60, 0.64] | 0.14 [0.09, 0.19] | 0.90 [0.90, 0.91] | −0.10 [-0.11, -0.08] |
| (1,4) | 0.57 [0.54, 0.59] | 0.13 [0.10, 0.15] | 0.76 [0.73, 0.80] | 0.00 [-0.02, 0.01] |
| (2,2) | 0.84 [0.79, 0.88] | 0.11 [0.09, 0.12] | 0.96 [0.95, 0.97] | 0.00 [-0.01, 0.00] |
| (2,3) | 0.86 [0.84, 0.89] | 0.10 [0.08, 0.13] | 0.95 [0.95, 0.97] | −0.04 [-0.06, -0.01] |
| (2,8) | 0.85 [0.80, 0.90] | 0.11 [0.09, 0.12] | 0.98 [0.97, 0.98] | 0.01 [-0.03, 0.03] |
| (3,6) | 0.78 [0.72, 0.85] | 0.11 [0.08, 0.12] | 0.98 [0.96, 0.98] | −0.03 [-0.05, -0.01] |
| (4,3) | 0.58 [0.55, 0.63] | 0.32 [0.23, 0.39] | 0.90 [0.88, 0.93] | −0.15 [-0.19, -0.12] |
| (7,1) | 0.82 [0.79, 0.84] | 0.10 [0.07, 0.12] | 0.87 [0.85, 0.88] | −0.02 [-0.05, 0.01] |
| (7,9) | 0.66 [0.63, 0.68] | 0.24 [0.21, 0.28] | 0.90 [0.88, 0.93] | −0.04 [-0.05, -0.02] |
| (8,0) | 0.85 [0.81, 0.87] | 0.11 [0.07, 0.16] | 0.98 [0.96, 0.98] | −0.08 [-0.12, -0.05] |
| (8,2) | 0.75 [0.72, 0.79] | 0.19 [0.15, 0.23] | 0.94 [0.93, 0.96] | −0.07 [-0.09, -0.04] |
| (8,4) | 0.80 [0.80, 0.81] | 0.12 [0.09, 0.16] | 0.90 [0.87, 0.92] | 0.02 [0.00, 0.04] |
| (8,6) | 0.77 [0.75, 0.78] | 0.15 [0.11, 0.18] | 0.91 [0.85, 0.93] | −0.03 [-0.05, -0.01] |

RL agents can only access the current environment at any time period, it is more compatible with the online learning required by the majority of RL algorithms.

We calculate the final performance and forgetting for the first task in all task sequences at the end of the whole training, and the results are summarized in Table 6. It can be seen that Perfect Memory can prevent forgetting well on the sequence of RL tasks after modifying the replay mode from offline to online. Therefore, we can conclude that the forgetting of Perfect Memory discussed above is primarily caused by the offline learning of the replayed tasks.

### C.3 Parameter Analysis

We vary the coefficient for policy distillation in RECALL and measure its impact on the three evaluation metrics: average performance, forgetting, and forward transfer. We run experiments on the first task sequence of CW6. The results are presented in Table 7, indicating that appropriate policy distillation of the actor can significantly improve performance.

It is worth noting that we exclusively conduct policy distillation for actor while not for critic in RECALL. This is because, regardless of critic, the agent interacts with the environment to collect data just by carrying out the policy from actor, and the primary goal of our policy distillation mechanism is to reduce distributional

Table 7: Average performance, forgetting, and forward transfer metrics on the first task sequence of CW6 for RECALL, for different values of the policy distillation regularization coefficient $\lambda$.

| Regularization Coefficient $\lambda$ | Ave. Perf. | Forgetting | F. Transfer |
|---|---|---|---|
| 0.01 | 0.84 [0.82, 0.86] | 0.02 [0.01, 0.04] | 0.37 [0.34, 0.39] |
| 0.1 | 0.84 [0.82, 0.87] | 0.01 [0.00, 0.02] | 0.39 [0.38, 0.40] |
| 1 | 0.87 [0.86, 0.88] | −0.01 [-0.01, 0.00] | 0.34 [0.33, 0.36] |
| 10 | 0.95 [0.94, 0.95] | −0.05 [-0.06, -0.04] | 0.36 [0.32, 0.40] |
| 100 | 0.88 [0.86, 0.89] | −0.05 [-0.06, -0.03] | 0.18 [0.16, 0.20] |

shift between the dataset and the learned policies caused by offline learning on old tasks, that is, the deviation between the replayed experience distribution of old tasks and their action policies.

## C.4  Performance Curves results

We also provide the average performance curves in CW6 task sequences for RECALL and baselines, as shown in Figure 10.

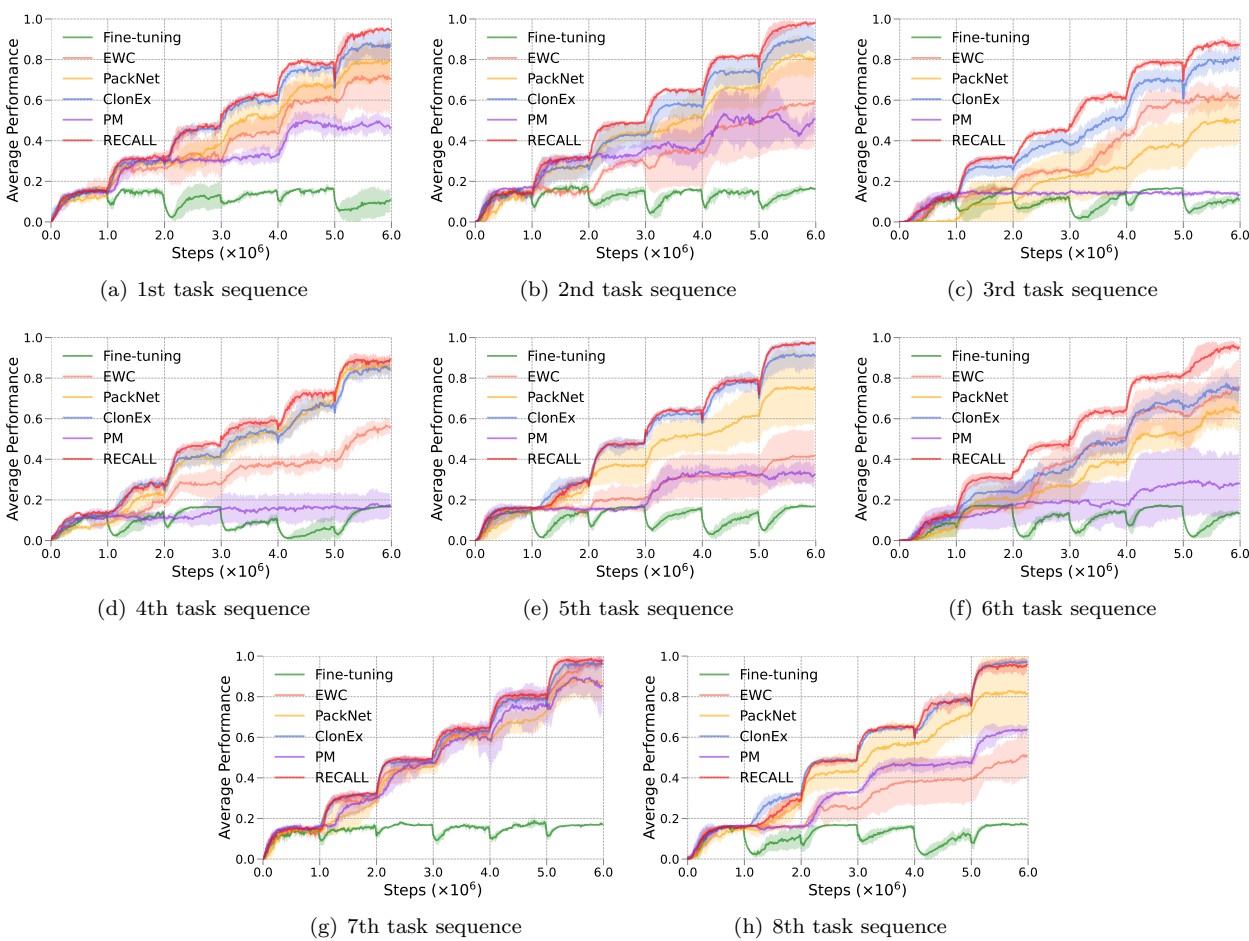

Figure 10: Average (over tasks) success rate for all methods in CW6 task sequences.

Table 8: Summary of the additional computation steps required by all methods in addition to the original SAC training computation. "-" indicates that the corresponding method does not involve that step.

| Method | Regularization Term | Additional Training |
|---|---|---|
| Fine-tuning | - | - |
| EWC | weight consolidation | - |
| PackNet | - | retraining parts of the network after pruning |
| ClonEx | behavioural cloning | - |
| Perfect Memory | - | - |
| RECALL | policy distillation | PopArt parameters updates |

Table 9: Statistics of the data storage in the training process of all methods, where $N$ is the number of tasks in the task sequence to be learned. "-" indicates that the corresponding method does not involve that term.

| Method | $\mathcal{D}_{new}$ size | $\mathcal{D}_{old}$ size |
|---|---|---|
| Fine-tuning | | - |
| EWC | | - |
| PackNet | $1 \times 10^6$ | - |
| ClonEx | | $N \times 10^4$ |
| Perfect Memory | | $(N-1) \times 10^6$ |
| RECALL | | $(N-1) \times 10^6$ |

# D    Computational Efficiency and Data storage

To help readers further understand the learning process of RECALL and all baselines, we summarized the major additional computation steps involved in all methods in addition to the original SAC training computation as well as the comparison of replay buffer sizes required by each method. The results are shown in Tables 8 and 9, respectively.

Fine-tuning and Perfect Memory are the two simplest methods, and both require exactly the same amount of computation as the original SAC training. The key difference between them is that Perfect Memory needs to store data of historical tasks and replay them while learning new tasks, while FT only stores and learns current task experiences at any time. EWC and ClonEx are regularization-based methods, wherein EWC involves Fisher information matrix calculation for estimating the importance of neural network weights, and ClonEx needs to store historical data for behavioral cloning. In contrast, PackNet, a parameter isolation method, gets rid of the above constraints of computing regularization terms and storing historical experiences, but it introduces additional retraining of parts of the network at task change.

Our proposed method, RECALL, is a method that combines experience replay and regularization constraints. Compared to Perfect Memory, although it introduces additional computation related to policy distillation and target normalization, the resulting performance improvement is quite significant. Compared to regularization-based methods such as EWC and ClonEx, RECALL introduces additional parameters update about the target normalization. Nevertheless, the number of parameters in this part is relatively small compared to the massive weights in the actor and critic networks common in all methods. In terms of historical data storage, as discussed in our related work in the main text, there have been a lot of studies that show that preserving a small amount of selective experiences or reconstructing observations through a generative model as an alternative is also sufficient.

