# OpenReview forum: "Replay-enhanced Continual Reinforcement Learning"
_TMLR — Accepted by TMLR_

### Review · Reviewer_ZCap · 2023-08-05

**Summary Of Contributions:**

The authors introduce RECALL, a new method for continual reinforcement learning which combines experience replay with policy distillation to combat forgetting and target normalization to encourage forward transfer. This method is evaluated on the Continual World benchmark against a suite of benchmarks, and often out performs all of them. The two components of RECALL (policy distillation and target normalization) are then ablated, shows that both contributions are relevant for improving performance.

**Audience:**

Yes

**Claims And Evidence:**

Yes

**Requested Changes:**

These directly correspond to my 3 weaknesses, plus one observation.

1) Show that the backwards transfer your baseline (i.e. perfect memory) is actually insufficient (e.g. maybe the drop is small in aggregate, but corresponds to a small number of dramatic failures, or maybe less replay improves forward transfer at the cost of backwards transfer) or reframe the paper around preserving backwards transfer while addressing the clear failure of forwards transfer.

2) Either add additional analyses to back up the strong claims of your method's superiority, or remove the offending highlights and hedge your claims.

3) Provide evidence that changes in reward magnitude correlate with changes in forward transfer and that changes in off-policyness (e.g. importance sampling ratios) correlate with changes in backwards transfer.

4) (Optional) Your policy distillation loss + RL loss is quite similar to CQL for offline RL. Since older tasks are functionally offline RL problems in your setup, this provides a slightly different motivation. If you rerun with CQL and it still works, I'd recommend going with that (which also obviates the need to provide the evidence I asked for in requested change 3).

**Strengths And Weaknesses:**

# Strengths

## Paper is clearly written and easy to follow

I really appreciate how easy it was to identify the contributions and see what specific problems they were designed to overcome.

## Experiments were well thought out

Qualitative results to motivate the problem, followed by a broad comparison to alternative methods, and proper ablations -- well done!

# Weaknesses

## Is forgetting still an issue for experience replay based methods?

You claim backwards transfer / catastrophic forgetting is a key problem you want to address, and then demonstrate that it is a problem in Figure 1. But this figures shows an average degradation in the single digits percentage points whereas forward transfer failures are seen to be massive (literally 1 to 0 in one case). It would've been fine to focus the paper around preserving the already adequate backwards transfer of experience replay while improving its forward transfer. Alternatively, you should demonstrate that these very small drops are behaviorally significant or find a problem setting where they are more pronounced.

## Results interpreted as being stronger than the given analysis warrants

You are doing better than most by providing a reasonable measure of spread (90% bootstrapped CI) in your results figures. But then you proceed to highlight the 'best' result in cases where the CIs significantly overlap. I understand the tendency -- I imagine many of these results might become significant with extra seeds (which might be prohibitive to run given the breadth of your baselines). I'd recommend some of the methods found in [1] that would allow you to better express significance in cases where you run many related experiments (e.g. CW6 tasks 1-8). Barring these new analyses, I'd recommend simply removing the offending highlights and hedging your language somewhat.

## The motivation for your method relies on mechanisms that are never shown empirically

You claim that forward transfer is poor because of the change in reward magnitudes that occur when going between tasks and backwards transfer is poor because of off-policy updates, but neither are shown empirically. This should be straightforward to address: just show that reward magnitudes change across task boundaries and (ideally) that the magnitude of the change correlates with a breakdown in forward transfer (and do the analogous thing for the backwards transfer/off policy case).

A small related point -- shouldn't something like the TD-error magnitude matter more than the raw reward magnitude? And I would imagine this would actually be larger for the new task, since (at convergence) these errors would go to zero.


[1]: Deep Reinforcement Learning at the Edge of the Statistical Precipice

---

> ### Author Response · Authors · 2023-09-19
> **Reply to Reviewer ZCap**
>
> > Q1: Is forgetting still an issue for experience replay based methods? Show that the backwards transfer your baseline (i.e. perfect memory) is actually insufficient (e.g. maybe the drop is small in aggregate, but corresponds to a small number of dramatic failures, or maybe less replay improves forward transfer at the cost of backwards transfer) or reframe the paper around preserving backwards transfer while addressing the clear failure of forwards transfer.
>
> Although experience replay based methods can largely prevent forgetting by continuing to train the model on the experiences of historical tasks, forgetting is still a non-negligible problem for this kind of methods in continual RL tasks. This is mainly due to the fact that when learning a new task, the experience replay of old tasks is a kind of offline learning based on historical data, rather than online learning as expected by the majority of RL algorithms. This offline learning mode may cause standard off-policy RL methods fail due to overestimation of values induced by the distributional shift between the dataset and the learned policy, which in turn degrades the agent's well learned performance on previous tasks and results in forgetting. This is different from supervised learning that is essentially a form of offline learning.
>
> We apologize for any possible confusion caused by our unclear expression. In our revised manuscript, we followed the reviewer's suggestion and visualized the learning curves of the first task in the 13 pairs of task sequences with more than 0.1 forgetting, to clearly demonstrate that forgetting in RL task sequence is still a significant issue for experience replay based methods. Please refer to the first half of Appendix C.2 (Figure 9) for more experimental results details.
>
>
> > Q2: Either add additional analyses to back up the strong claims of your method's superiority, or remove the offending highlights and hedge your claims.
>
> Thanks to the reviewer for pointing out our inappropriate highlighting of some experimental results. In the revised version, we have corrected the bold representation of the best results in Tables 1, 2, and 3, and revised the relevant content of the abstract, experiments, and conclusions sections to hedge our claims.

---

> ### Author Response · Authors · 2023-09-19
> **Reply to Reviewer ZCap**
>
> > Q3: Provide evidence that changes in reward magnitude correlate with changes in forward transfer and that changes in off-policyness (e.g. importance sampling ratios) correlate with changes in backwards transfer.
>
> Thanks for the reviewer's suggestion. In the revised manuscript, we added some additional experiments to strengthen and improve the relevant content in the following two aspects:
> - *Plasticity analysis of Perfect Memory:* Concerning the effect of reward scale on forward transfer, we find it difficult to undertake direct experimental demonstration by intentionally regulating different magnitudes of rewards, because reward in a complex setting involves many environmental dynamics aspects that are difficult to change artificially. Nonetheless, we attempted to present some circumstantial evidence on this subject from some other perspectives. We added the details in Appendix C.1 of our revised manuscript.
>     - Firstly, we added some visualizations of the losses and Q value learning curves on example task sequence where Perfect Memory fails, further demonstrating that Perfect Memory does not achieve any effective model updates on new tasks (see Figure 8).
>     - Secondly, for Perfect Memory, we conducted additional experiments with four combinations of actor and critic, in which the actor and critic networks were either shared or not among tasks. The results (see Table 5) demonstrate that the limited plasticity suffered by Perfect Memory is primarily due to the shared critic network, that is, there is something wrong with the optimization of the shared value function.
>     - Finally, inspired by the value function scaling problem caused by the difference in reward scale between tasks discussed in multi-task RL, we analyze that the aforementioned limited plasticity of PM may also be produced by this reason. Concretely, in replay-based continual RL, we need to learn value functions on both current and past task experiences. Since each task generally have different reward scales, the output range of value functions applicable to each task may also differ, leading to a distraction dilemma. In continual learning, it is generally assumed that the model has achieved good performance on the old task when learning a new task. Therefore, the distraction dilemma is manifested as that the model continues to pay attention to the old task that has received larger rewards and completely ignores the learning of the new task with smaller initial observed rewards, which is consistent with the phenomenon of limited plasticity of Perfect Memory in our work.
> In addition, we propose to solve this issue from reward scale perspective, and extensive experimental results provided in our manuscript also in turn prove its effectiveness.
>
>     For the above analysis, we also added corresponding supplements in the paragraph ``Limited plasticity'' of Section 3 to further enrich and improve our manuscript.
> - *Forgetting Analysis of Perfect Memory:* Inspired by the fourth question from the reviewer, we acknowledge that the cause of forgetting mentioned in our work should be more accurately described as offline learning of historical tasks rather than off-policy learning. We have revised all statements in this regard in our newly submitted manuscript. In addition, we verified that the forgetting shown in our work is indeed essentially a result of that cause by conducting further experiments of Perfect Memory using offline and online replay modes, respectively. Please see Table 6 and related analysis in Appendix C.2 for more detailed information.

---

> ### Author Response · Authors · 2023-09-19
> **Reply to Reviewer ZCap**
>
> > Q4: (Optional) Your policy distillation loss + RL loss is quite similar to CQL for offline RL. Since older tasks are functionally offline RL problems in your setup, this provides a slightly different motivation. If you rerun with CQL and it still works, I'd recommend going with that (which also obviates the need to provide the evidence I asked for in requested change 3).
>
> Thanks to the reviewer's suggestion. We revisited our approach and acknowledged that the cause of forgetting mentioned in our work is more accurately described as offline learning of historical tasks rather than off-policy learning. We have revised all statements in this regard in our newly submitted manuscript.
>
> In addition, we believe that replacing the policy distillation in our algorithm with CQL to facilitate the agent's offline learning of historical tasks should theoretically be effective. However, for the problem we studied, policy distillation is a simpler and more straightforward technique, and we have also proved experimentally that it is quite effective. We continue to employ this technique in the most recent manuscript submission for the following reasons:
> - On the one hand, effective algorithms in continual RL have generally learned appropriate policies on past tasks before learning a new task. As a result, to avoid the distribution drift caused by offline learning on old tasks, we can directly use the policy distillation technique to keep the current learned policy on old tasks consistent with the approximate optimal policy learned at the end of the corresponding task learning period, which has been proven effective in our experiments and is relatively easy to implement in practice.
> - On the other hand, the offline RL method CQL, seeks to address the aforementioned restrictions by learning a conservative Q-function, which means that the model needs be totally relearned on previous tasks. This is a relatively tedious and complex process, and the outcomes may be inferior to those obtained by policy distillation.

---

### Review · Reviewer_qBJi · 2023-08-07

**Summary Of Contributions:**

The primary contribution of this paper lays at what they call RECALL, which is a set of modifications to using replay as a means to overcome the plasticity/catastrophic forgetting issues faced in continual reinforcement learning. The modifications consist of two parts: 1) The use of PopArt to enforce scale invariance for the set of value-based RL policies learned in their network, and 2) a policy distillation loss function to mitigate forgetting caused by the off-policy nature of replaying old tasks. They set out to verify their modifications improve the performance of replay based and SOTA based approaches for continual reinforcement learning. The empirical section consists of a series of results in Continual World, starting with a controlled experiment only looking at two sequential tasks at a time and then moving towards more difficult variants of CW6, CW10, and CW20. RECALL does admirably, outpacing both perfect replay and the set of SOTA algorithms.

**Audience:**

Yes

**Claims And Evidence:**

Yes

**Requested Changes:**

Overall, I think the manuscript is clean and written well. I believe the above weaknesses should be easily rectified through some editing and re-motivating the work from the perspective of multi-tasking reinforcement learning (at least clarifying this as such) (**W1**), and clarifying how their approach uses some modification from ClonEx (**W3**) and how your approach may have more data than others (**W4**). While 1,3,4 are most important in my eyes, the other weaknesses should also be addressed to the best of the ability of the authors.

**Strengths And Weaknesses:**

Overall, the paper is well written and the method is well tested empirically in continual world. The method is also, mostly, well discussed and presented with only a few lingering questions (see the weaknesses). While I have many more weaknesses than strengths, they are mostly effort the authors need to take to clarify some aspects of their work or better motivate their work in clearer terms.


W1. While the main motivation of the paper is rooted in continual learning, many limitations to the method are apparent in the case of continual (lifelong) RL as defined by (Khetrapal et al., 2022). Instead, the focus of this work seems to more aptly follow from multi-task reinforcement learning. This distinction should be made clear, especially because of a lack of discussion around making this method work for continual (lifelong) RL.

W2. For the distill loss, it is unclear where $\pi_{old}$ comes from. Is this data stored along with the experience, or is it re-computed in some way? This should be clarified.

W3. While this method is introduced in isolation, it seems like several innovations from ClonEx are employed (as discussed in section 5.1 in the implementations details). While I appreciate the details, I feel as though these should be included in your method specification and possibly as part of the ablation study (if that is in your computational budget). With this weakness, I find myself uncertain what algorithms is being run labeled *None* in the ablation study. Is it closer to ClonEx, or something else?

W4. Do all the baselines sample from all the data generated throughout the agent's lifetime? If not, this should be clarified. In any case, you might want to highlight that this is a limitation of all the methods you propose, as it is a significant limitation of this (and any "perfect memory" approach) approach for C(LL)RL.

W5. You do not include how hyperparameters were selected, or the empirical analysis of the regularization coefficient for policy distillation in the main paper. While these results are in the appendix, I think they should be discussed in the main paper. At the very least there should be a statement that you are using the hyperparameters from (Wolczyk et al., 2022)

W6. You should use (Khetrapal et al., 2022) to relate your work to the reset of CRL. There are lots of definitions pervasive, and any new work should better understand which specific problem they are solving, not letting the motivation of the general C(LL)RL encroach into different, more specific problems (at least not unknowingly).

**Other misc Questions:**
- Q1. In figure 1(b), why do you think there are poor results across the diagonal? Is this to do with the fact we are learning a new policy head and the initialization may not suite the current representation of the network?

Edits:
- I think the caption for Figure 1 is slightly confusing because of the use of the proper task titles ("representing the learning of task sequence M=[PUSH-V1, HAMMER-v1]").

---

> ### Author Response · Authors · 2023-09-19
> **Reply to Reviewer qBJi**
>
> > Q1: While the main motivation of the paper is rooted in continual learning, many limitations to the method are apparent in the case of continual (lifelong) RL as defined by (Khetrapal et al., 2022). Instead, the focus of this work seems to more aptly follow from multi-task reinforcement learning. This distinction should be made clear, especially because of a lack of discussion around making this method work for continual (lifelong) RL.
>
> The main motivation of this paper is to address the problem mentioned in (Wołczyk et al., 2021; 2022) that the experience replay based methods (such as Perfect Memory) have been shown in numerous studies to consistently perform well in supervised continual learning, but perform rather poorly on the latest continual RL benchmark Continual World.
>
> By systematically conducting some preliminary experiments (see relevant discussion and experimental results in Section 3 and Appendices C.1 and C.2), we found that the problem is mainly manifested in limited plasticity and a certain degree of forgetting, and the possible reasons for this result are the value function scaling problem caused by different reward scales, and the offline learning on replayed tasks.
>
> For the former, we try to integrate PopArt normalization technique into replay-based continual learning algorithm to handle it, mainly for the following reasons: The PopArt method is a technique specifically used to address the value function scaling problem in deep RL (van Hasselt et al., 2016). It scales the value function to ensure that its output is within a suitable range, which helps to improve training stability and efficiency. In multi-task RL, agents typically switch between different tasks, hence the need to train different value functions for each task. Since each task may have different reward signals, the output range of each task’s value function may also differ. This leads to the value function scaling problem (Hessel et al., 2019). Similarly, in replay-based continual RL, value functions on both current and past task experiences need to be learned while learning a new task. Since each task also generally have different reward scales, we can view this process as a multi-task learning on current and historical tasks. Therefore, applying the PopArt method to address the value function scaling problem in replay-based continual RL is reasonable and necessary.
>
> Thanks for the suggestion. We rephrased our motivation (see Section 1) and added the above discussion around making this method work for continual RL (see Section 4) in our revised manuscript.
>
> > Q2: For the distill loss, it is unclear where $\pi_{old}$ comes from. Is this data stored along with the experience, or is it re-computed in some way? This should be clarified.
>
> We apologize for any possible confusion caused by our unclear expression. Here, $\pi_{old}$ is the historical policy obtained after ending the training on the associated replayed task. In our implementations, before each new task training starts, we compute $\pi_{old}(\cdot|s_t)$ for all experiences of the previous task through the latest learned policy and store them along with the corresponding experiences in $\mathcal{D}_{old}$ for subsequent use. In our revised manuscript, we followed the suggestion and clarified this in Section 4. Please see the paragraph below Equation 8. Thanks.
>
> **Reference**
>
> - Wołczyk et al. Continual world: A robotic benchmark for continual reinforcement learning. In Advances in Neural Information Processing Systems, volume 34, pp. 28496–28510, 2021.
>
> - Wołczyk et al. Disentangling transfer in continual reinforcement learning. In Advances in Neural Information Processing Systems, volume 35, pp. 6304–6317, 2022.
>
> - van Hasselt et al. Learning values across many orders of magnitude. In Advances in Neural Information Processing Systems, volume 29, 2016.
>
> - Hessel et al. Multi-task deep reinforcement learning with popart. In Proceedings of the AAAI Conference on Artificial
> Intelligence, volume 33, pp. 3796–3803, 2019.

---

> ### Author Response · Authors · 2023-09-19
> **Reply to Reviewer qBJi**
>
> > Q3: While this method is introduced in isolation, it seems like several innovations from ClonEx are employed (as discussed in section 5.1 in the implementations details). While I appreciate the details, I feel as though these should be included in your method specification and possibly as part of the ablation study (if that is in your computational budget). With this weakness, I find myself uncertain what algorithms is being run labeled None in the ablation study. Is it closer to ClonEx, or something else?
>
> We apologize for any possible misunderstanding caused by our unclear expression. Here, it should be clarified that:
> - Our proposed method RECALL and ClonEx are two different classes of continuous learning methods, the main difference between them is that they use completely different mechanisms to mitigate catastrophic forgetting. To be specific, ClonEx is a regularization-based method that reduces forgetting by adding a behavioral cloning regularization term to constrain updates of network weights, while RECALL maintains the training on past tasks by replaying experiences while learning new tasks to avoid forgetting. We have included a detailed comparative analysis of them in the second and third paragraphs of section 5.3 in our original manuscript.
> - As for the implementation details in Section 5.1, we employ the best-return exploration in RECALL to facilitate exploration when the new task begins, as used by ClonEx, which is only a way of initial exploration for the agent in the new task, and can also be regard as a way of initializing the network output head corresponding to the new task. This way has been proven to be effective in promoting forward transfer in (Wołczyk et al., 2022) and can be used similarly in all continual RL algorithms using multi-head networks. In order to ensure the fairness of the comparison, we also adopted this mechanism in RECALL, which is not an innovation. Nevertheless, thanks to the reviewer's suggestion, and we have added this detail to our method description as well. Please refer to the last paragraph of Section 4 and line 7 of the Algorithm 1 pseudo-code in our revised manuscript.
> - In the ablation study, we mainly consider the individual effects of our proposed target normalization and policy distillation mechanisms on the naive experience replay method. The algorithm labeled as None means either target normalization nor policy distillation mechanism is used, i.e., degenerating to the naive experience replay with best-return exploration. We restated this in section 5.4 of the revised manuscript.
>
> > Q4: Do all the baselines sample from all the data generated throughout the agent's lifetime? If not, this should be clarified. In any case, you might want to highlight that this is a limitation of all the methods you propose, as it is a significant limitation of this (and any "perfect memory" approach) approach for C(LL)RL.
>
> As we introduced in the paragraph ``Baselines'' of section 5.1, the baseline method we compare in our experiments include Fine-tuning without considering any forgetting, the regularization-based methods EWC and ClonEx, the parameter isolation method PackNet, and the experience replay method Perfect Memory. Like our proposed RECALL, replay-based methods such as Perfect Memory need to store all or partly selective historical data for replaying. ClonEx also needs to store part of the historical data for behavioral cloning, and other several methods do not need to store the historical data during the training process.
>
> We followed the reviewer's suggestion and added the comparison of all baselines as well as our method in terms of data storage in the revised manuscript. Please see Appendix D (Table 9) for more details. Thanks.
>
> > Q5: You do not include how hyperparameters were selected, or the empirical analysis of the regularization coefficient for policy distillation in the main paper. While these results are in the appendix, I think they should be discussed in the main paper. At the very least there should be a statement that you are using the hyperparameters from (Wolczyk et al., 2022).
>
> Thanks for the suggestion. We have moved the statement about the selection of the regularization coefficient $\lambda$ for policy distillation in RECALL from appendix to the paragraph ``Implementations'' of Section 5.1 in the main paper.

---

> ### Author Response · Authors · 2023-09-19
> **Reply to Reviewer qBJi**
>
> > Q6: You should use (Khetrapal et al., 2022) to relate your work to the reset of CRL. There are lots of definitions pervasive, and any new work should better understand which specific problem they are solving, not letting the motivation of the general C(LL)RL encroach into different, more specific problems (at least not unknowingly).
>
> We followed the suggestion and rephrased as well as clarified the specific problem that our work addresses, in conjunction with the work in (Wolczyk et al., 2022). Please refer to Section 1 in our revised manuscript for details. Thanks.
>
> > Q7: (Other misc Questions) In figure 1(b), why do you think there are poor results across the diagonal? Is this to do with the fact we are learning a new policy head and the initialization may not suite the current representation of the network?
>
> Regarding the reason for some poor results that also exist on the diagonal in Figure 1(b), our analysis is as follows: Although the corresponding first and second tasks on the diagonal are identical, as the reviewer said, in our experiments, a new policy head is learned for the new task (i.e., the second task) by default for all pairwise task sequences. That is to say, we uniformly treat all the second tasks as a new task in their corresponding task sequence. Since the magnitude and density of the rewards may differ dramatically at various learning phrases with the same task, the task sequences on the diagonal are also likely to face poor plasticity like the task sequences at other positions suffered. However, since the two tasks are identical, i.e., the shared representation learned on the first task also applies to the second task, the final performance obtained by the model on the second task on the diagonal is better than that on the other task sequences in the same column.
>
> > Q8: (Edits) I think the caption for Figure 1 is slightly confusing because of the use of the proper task titles ("representing the learning of task sequence M=[PUSH-V1, HAMMER-v1]").
>
> In Figure 1, each grid of the evaluation matrices represents a specific pairwise task sequence. The numbers $0\sim9$ indicate identifications of ten different tasks contained in Continual World benchmark, and the mapping between them and the proper task names are shown in Figure 7 in Appendix A. In conjunction with Figure 7, we gave an example in the caption of Figure 1, to make the readers more clearly understand the specific names of the sequential tasks corresponding to each grid. That is, if the identifications of two tasks is 6 and 0 respectively, which represents task sequence $\mathcal{M}=[$PUSH-V1, HAMMER-V1]; While if the identifications of two tasks is 5 and 0 respectively, then the specific task sequence is $\mathcal{M}=[$HANDLE-PRESS-SIDE-V1, HAMMER-V1]. We hope this explanation will clear up your confusion, and we have also rephrased and clarified the relevant content in our revised manuscript. Thanks.

---

### Review · Reviewer_6uKu · 2023-09-06

**Summary Of Contributions:**

The authors present a method for lifelong reinforcement learning which combines a large replay buffer with pop-art normalization. The main contribution of this paper is to show that plasticity issues that have been pointed out to hold back replay methods in prior work can be explained by loss normalization issues and addressed with corresponding methods that have previously been proposed in the multi-task reinforcement learning literature. The authors evaluate the method on a recently proposed benchmark of continual robotics tasks and compare against similar methods as have been tested on that benchmark before. The authors furthermore evaluate an additional regularization term, constraining the KL divergence to prior policies.

**Audience:**

Yes

**Broader Impact Concerns:**

No concerns.

**Claims And Evidence:**

Yes

**Requested Changes:**

+ Please add additional discussion regarding the limitations of the multi-head approach as well as how this work relates to plasticity issues observed in supervised learning.
+ If possible, please add value/loss/gradient magnitude or similar curves to illustrate the problem with perfect memory.
+ Is there supporting evidence that can be added that the issues ascribed to off-policyness are indeed due to that reason?
+ Could you clarify how much of the perceived plasticity issues with the baseline approach lies in the critic compared to the policy? The suggested solution suggests this is mainly an issue of the critic but showing this would strengthen the claim. Reinitializing the corresponding networks between tasks may show this.

**Strengths And Weaknesses:**

At a first glance, the method itself is not particularly novel as it combines two methods that address orthogonal issues. However, it effectively addresses issues raised in prior work and achieves good results with simple methods. The evaluation is generally extensive enough to support the most important claims. As a result, the paper should be of interest to the lifelong RL community.

Strengths:
+ The paper is well written and easy to read.
+ The proposed method is effective and tested on a previously established benchmark while also being simple and straightforward to apply.

Weaknesses:
+ One key strength of replay based approaches to continual learning is that they work well even when the task boundaries are unknown or + the dynamics are transitioning in a smooth way. The same is not true for pop-art, reducing the applicability of this method. The authors  claim that this method can be a drop-in replacement for replay based approaches but this claim should be moderated.
+ The authors show that in the case of this benchmark, the perceived lack of plasticity is actually due to issues of scaling. Advantages go up as the agent observes more reward and the scale of values increases and only drop later in training as the agent converges (the authors imply that they are highest for solved tasks but while this is possible, it should not be the case when the capacity of the network is sufficient); however, this is a phenomenon that is closely linked to RL algorithms while plasticity issues have been observed in supervised settings as well. In that case, the magnitude of the loss would be highest when the task is new and a similar normalization procedure would likely not work. The work could do more to identify when plasticity issues are due to scaling issues and when they are due to other reasons.
+ I’m not convinced that the claim of the effect of off-policyness is backed up by evidence in this work. The distillation loss can also be seen as a regularizer, similar to other regularizers in the continual learning literature and may help with plasticity more generally rather than simply addressing off-policyness.

---

> ### Author Response · Authors · 2023-09-19
> **Reply to Reviewer 6uKu**
>
> > Q1: Please add additional discussion regarding the limitations of the multi-head approach as well as how this work relates to plasticity issues observed in supervised learning.
>
> Regarding this question, we think it is necessary to clarify the following facts:
> - *Limitations of Multi-head Structure:* Although the use of multi-head network structure will weaken the advantage of replay-based methods that can still work well when task boundaries are unknown, there are numerous studies have shown that it can achieve significantly better results than single-head network architecture when task changes are known in continual learning scenarios, as experimentally demonstrated in (Wołczyk et al., 2022). Based on this fact, multi-head architecture is generally the first choice for continual learning algorithms when task boundaries are provided, as we do in our this work. To say the least, even if the task identifier is not provided, various existing automatic task identification techniques (like FMN (Milan et al., 2016)) can be utilized to somewhat compensate for this limitation. Thanks for the suggestion. We added discussion regarding above limitation of multi-head structure in the footnote 1 within Section 4.
> - *Plasticity Issue:* There are many reasons that can cause the issue of neural network plasticity. The issue described in our paper is a unique value function scaling problem in RL, which does not generally exist in supervised learning, especially classification tasks, due to the naturally well scaled loss functions (e.g., cross entropy), as we discussed in the third paragraph of Section 1 in our manuscript.
>
> > Q2: If possible, please add value/loss/gradient magnitude or similar curves to illustrate the problem with perfect memory.
>
> We followed the suggestion and conducted some visualizations of the actor and critic losses and Q value learning curves on example task sequence where Perfect Memory fails. From the results (see Figure 8), it can be observed that within a few time steps after the learning of a new task (i.e., the second task), the loss functions of actor and critic together with the output of the value function rapidly converge to a stable state that remains around 0, further indicating that the model of Perfect Memory method does not learn any useful information about the new task. Please refer to relevant content we added in Appendix C.1 in the revised manuscript. Thanks.
>
> > Q3: Is there supporting evidence that can be added that the issues ascribed to off-policyness are indeed due to that reason?
>
> In our revised manuscript, we revisited that the cause of forgetting mentioned in our work should be more accurately described as offline learning of historical tasks rather than off-policy learning. We have revised all statements in this regard in our newly submitted manuscript. In addition, we verified that the forgetting shown in our work is indeed essentially a result of that cause by conducting further experiments of Perfect Memory using offline and online replay modes, respectively. Please refer to Table 6 and related analysis in Appendix C.2 for more detailed information. Thanks.
>
> > Q4: The suggested solution suggests this is mainly an issue of the critic but showing this would strengthen the claim. Reinitializing the corresponding networks between tasks may show this.
>
> We followed the suggestion and conducted additional experiments for Perfect Memory with four combinations of actor and critic, in which the actor and critic networks were either shared or not shared among tasks. The results (see Table 5) demonstrate that the limited plasticity of the model on the new task suffered by Perfect Memory is largely caused by the shared critic network. The relevant content is added in Appendix C.1 in our revised manuscript. Thanks.
>
> **Reference**
>
> - Wołczyk et al. Disentangling transfer in continual reinforcement learning. In Advances in Neural Information Processing Systems, volume 35, pp. 6304–6317, 2022.
> - Milan et al. "The forget-me-not process." In Advances in Neural Information Processing Systems, volume 29, 2016.

---

### Review · Reviewer_FYXr · 2023-09-06

**Summary Of Contributions:**

This paper addresses the problem of continual RL, in which RL tasks are sequentially presented. The method presented ("RECALL") proposes two changes: Popart-style per-task normalization and policy distillation.

**Audience:**

Yes

**Broader Impact Concerns:**

No major concerns.

**Claims And Evidence:**

Yes

**Requested Changes:**

The below changes are listed in order of decreasing importance (ie first item most important).

- Address the interaction of SAC with Popart
- Add further experimental settings beyond robot manipulation.
- Add metrics that take into account compute and data storage expenses.

**Strengths And Weaknesses:**

## Strengths

The primary strength of the paper is performance of the method. The results consistently show that the proposed method has strong performance on the Continual World benchmark.

## Weaknesses

There are several limitations that should be addressed. First of all, we note that the method is a relatively simple combination of two established techniques (and the impact of policy distillation is fairly small). While the paper still has value, the methodological contribution of the paper is fairly weak. Other points:

- The authors do not meaningfully engage with the interaction between Popart/multi-task learning and SAC; in particular, different systems often require different target entropies or alpha coefficients for good performance. Meanwhile, the effective weight of the entropy in the loss function varies as both alpha and sigma vary. Perhaps the updates for alpha and sigma could be coupled to improve efficiency? In general, the investigation of this element of the methodology is non-existent and may potentially improve performance and strengthen the contribution of the paper.
- Have the authors considered adding the distillation loss to the SAC loss in place of the entropy term, corresponding to a history-based prior as opposed to a uniform prior? This is common in, for example, modern RLHF.
- The experiments are limited in the set of environments investigated. In particular, the authors only evaluation manipulation tasks, which may moderate the impact of varying entropy weights as discussed above. It would substantially strengthen the paper to include other evaluation settings.
- There are two potential limitations induced by the proposed changes. First, the Popart normalization may harm efficiency for small amounts of per-task data. Second, the distillation substantially increases training and data storage cost compared to (especially regularization-based) baselines. Ideally, metrics should be presented that report performance as a function of data and compute used. Currently, per-task data is fixed in experiments and the only discussion of compute requirements is the limited discussion in Section 5.5.

---

> ### Author Response · Authors · 2023-09-19
> **Reply to Reviewer FYXr**
>
> > Q1: Address the interaction of SAC with Popart (considering the effective weight of the entropy in the loss function).
>
> We fully agree with the analysis of the reviewer. In fact, we did take that into account in our experimental implementations.
> For our proposed method RECALL as well as all baselines in our experiments, we use an implementation of the underlying RL algorithm SAC based on (Wołczyk et al., 2021), in which the maximum entropy coefficient $\alpha$ is tuned/updated automatically according to the adjustment rule provided in (Haarnoja et al., 2018b). We really apologize for omitting this relevant description. In the revised manuscript, we added this details in the paragraph ``Implementations'' of section 5.1. Thanks.
>
> > Q2: Add further experimental settings beyond robot manipulation. (The experiments are limited in the set of environments investigated. In particular, the authors only evaluation manipulation tasks, which may moderate the impact of varying entropy weights as discussed above. It would substantially strengthen the paper to include other evaluation settings.)
>
> As mentioned in the reply to the reviewer's first question, we have already taken into account the impact of varying entropy weights in our experimental implementations by automatically adjusting the maximum entropy coefficient. We hope that this operation and clarification can dispel the reviewer's concerns regarding this aspect.
>
> As for the benchmark (i.e., Continual World) adopted in our experiments, it is a latest benchmark proposed in (Wołczyk et al., 2021) to be as a testbed for evaluating RL agents with respect to challenges incurred by the continual learning paradigm. All 10 tasks included in this benchmark were carefully selected from Meta-World, a suite of recognized benchmarks that has been already established as well as widely studied and used in multi-task and meta RL communities. Although the tasks included in this benchmark suite are all robotic manipulation tasks, they cover multiple domains, including robotic arm control, object manipulation, door/window control, and etc. Each task has different goals and reward functions, requiring agents to have different skills and abilities. In addition, the tasks in this benchmark have different difficulty levels, ranging from simple to complex, which can also help to evaluate the ability of algorithms to handle tasks of different difficulties.
>
> Returning to our work, the main motivation is to address the problem mentioned in (Wołczyk et al., 2021; 2022) that the experience replay based methods (such as Perfect Memory) perform rather poorly on Continual World. Therefore, following the experimental settings in (Wołczyk et al., 2021; 2022) and carrying out systematically evaluation of our method on this benchmark are required and technically sound. We will try to further test our method in some other continual RL environment settings in our future work. Thanks for your suggestion.
>
> > Q3: Add metrics that take into account compute and data storage expenses.
>
> Our proposed method, RECALL, is a method that combines experience replay and regularization constraints. Compared to Perfect Memory (naive experience replay method), although it introduces additional computation related to policy distillation and target normalization, the resulting performance improvement is quite significant as shown by our extensive experimental results. Compared to regularization-based methods such as EWC and ClonEx, RECALL introduces additional parameters update about the target normalization. Nevertheless, the number of parameters in this part is relatively small compared to the massive weights in the actor and critic networks common in all methods. In terms of historical data storage, as discussed in our related work in the main text, there have been a lot of studies that show that preserving a small amount of selective experiences or reconstructing observations through a generative model as an alternative is also sufficient.
>
> We followed the reviewer's suggestion and added the comparison of all baselines as well as our method in terms of computational efficiency and data storage in the revised manuscript. Please see Appendix D (includes Tables 8 and 9) for more details. Thanks.
>
> **Reference**
> - Wołczyk et al. Continual world: A robotic benchmark for continual reinforcement learning. In Advances in Neural Information Processing Systems, volume 34, pp. 28496–28510, 2021.
> - Wołczyk et al. Disentangling transfer in continual reinforcement learning. In Advances in Neural Information Processing Systems, volume 35, pp. 6304–6317, 2022.
> - Haarnoja et al. Soft actor-critic algorithms and applications. arXiv preprint arXiv:1812. 05905, 2018b.

---

> ### Author Response · Authors · 2023-09-22
> **Reply to Reviewer FYXr**
>
> > (Weakness 2) Have the authors considered adding the distillation loss to the SAC loss in place of the entropy term, corresponding to a history-based prior as opposed to a uniform prior? This is common in, for example, modern RLHF.
>
> This is an interesting idea to consider.
>
> In RLHF, the reinforcement learning process is the further fine-tuning of models learned from supervised pre-training, which are oriented towards the same or similar tasks. Here, the distillation loss (KL divergence term) is added to RL algorithm to penalize the RL policy from moving substantially away from the initial pre-trained model with each training batch, which can be useful to make sure the model outputs reasonably coherent text snippets. Without this penalty the optimization may start to generate text that is gibberish but fools the reward model to give a high reward, interfering with the model's efficient learning.
>
> In SAC, the maximum entropy loss is usually used to encourage exploration and avoid over-reliance on specific actions. If the sequential tasks in continual RL are highly correlated (similar to tasks in pre-training and fine-tuning in RLHF), then, in theory, adding the policy distillation loss of historical tasks to the SAC loss in place of the entropy term may help to transfer the prior knowledge from historical tasks to the current task to guide exploration and learning of the current task. However, in real-world settings, tasks might not be highly connected or even completely distinct. In other words, using the distillation loss based on historical priors as a replacement for the entropy term in this situation may not achieve the desired result. It may even have the opposite effect of limiting exploration and suppressing learning on the current task.
>
> We think that this is an interesting but complex problem, and the effectiveness of this approach would depend on various factors, including the specific task and environment being considered, as well as the design of the neural network architecture and training procedure. We intend to further investigate it in our future work. Thanks.

---

### Author Response · Authors · 2023-09-19
**Response to all reviewers (Major Changes)**

We thank the editor and the reviewers for their valuable and detailed feedback. Following suggestions of the editor and reviewers, in the revised manuscript, we (A) rephrased some inappropriate statements and clarified some unclear and possibly confusing expressions, (B) added additional discussions of the relevant content mentioned by the reviewers, and (C) conducted additional experiments to cover more performance comparisons and analysis.

To (A): Based on the suggestions of reviewer ZCap (Q2, Q4), we revised some inappropriate highlights in our experimental results and hedged some relevant claims, while also rephrased the cause of forgetting. We also followed the suggestions and confusions raised by reviewer qBJi (Q2, Q3, Q6), and clearly added some descriptions about the implementation of $\pi_{old}$, several experimental setting used in ClonEx, and the specific problem we aim to address.

To (B): Following the suggestions of reviewer qBJi (Q1, Q5), we added additional analysis and discussion about the relationship between our work and multi-task RL in terms of target normalization, and moved the description of hyperparameter $\lambda$ selection from appendix to the main paper. We also added additional discussion regarding the limitations of the multi-head approach according to the suggestion of reviewer 6uKu (Q1) and highlighted automating entropy adjustment for maximum entropy in SAC referring to analysis from reviewer FYXr (Q1).

To (C): Based on the comments of four reviewers, we conducted additional experimental comparisons and analysis in the following aspects: (1) Inspired by the points of reviewer ZCap (Q1,Q3) and reviewer 6uKu (Q2,Q3,Q4), we conducted additional learning curves visualization and experimental analysis on plasticity and forgetting issues in Perfect Memory, as well as further explored and validated the possible causes of them (See Appendices C.1 and C.2). (2) In combination with the suggestions of reviewer qBJi (Q4) and reviewer FYXr (Q3), we added qualitative and quantitative comparisons with all baselines in terms of computational efficiency and data storage (See Appendix D).

For the convenience of reviewers, we reiterate all reviewer comments below. Furthermore, we provide an extra colored version of the resubmitted manuscript in which the changes are highlighted as blue text. For newly constructed figures and tables, we only colored the figure or table captions. The version with the text highlights is otherwise identical to the PDF of the resubmitted manuscript and appendix.

---

### Decision · Action_Editor_2kxh · 2023-10-27

**Recommendation:** Accept as is

**Comment:**

Although the reviewers agreed that this paper was a combination of pre-existing methods, they also all agreed that the results were solid and the experimentation was thorough.
The reviewers also all appreciated the authors' revisions of the paper based on their feedback.
All reviewers were ultimately supportive of accepting this work, and I agree with them.

**Audience:**

Yes.

**Claims And Evidence:**

The method proposed in this paper was thoroughly evaluated and yielded strong performance, which was the main strength agreed on by all reviewers.
The paper, especially after revisions, is well-written and clear.